# maxATAC: Genome-scale transcription-factor binding prediction from ATAC-seq with deep neural networks

**Tareian A. Cazares**[1], **Faiz W. Rizvi**[2], **Balaji Iyer**[3,4], **Xiaoting Chen**[5], **Michael Kotliar**[6], **Anthony T. Bejjani**[7], **Joseph A. Wayman**[8], **Omer Donmez**[5], **Benjamin Wronowski**[6], **Sreeja Parameswaran**[5], **Leah C. Kottyan**[5,9,10], **Artem Barski**[6,9,10], **Matthew T. Weirauch**[3,5,9,10,11], **V. B. Surya Prasath**[3,4,9], **Emily R. Miraldi**[3,4,8,9]*

1 Immunology Graduate Program, University of Cincinnati College of Medicine, Cincinnati, Ohio, United States of America, 2 Systems Biology and Physiology Graduate Program, University of Cincinnati College of Medicine, Cincinnati, Ohio, United States of America, 3 Division of Biomedical Informatics, Cincinnati Children's Hospital Medical Center, Cincinnati, Ohio, United States of America, 4 Department of Electrical Engineering and Computer Science, University of Cincinnati, Cincinnati, Ohio, United States of America, 5 The Center for Autoimmune Genetics and Etiology (CAGE), Cincinnati Children's Hospital Medical Center, Cincinnati, Ohio, United States of America, 6 Division of Allergy and Immunology, Cincinnati Children's Hospital Medical Center, Cincinnati, Ohio, United States of America, 7 Molecular and Developmental Biology Graduate Program, University of Cincinnati College of Medicine, Cincinnati, Ohio, United States of America, 8 Division of Immunobiology, Cincinnati Children's Hospital Medical Center, Cincinnati, Ohio, United States of America, 9 Department of Pediatrics, University of Cincinnati College of Medicine, Cincinnati, Ohio, United States of America, 10 Division of Human Genetics, Cincinnati Children's Hospital Medical Center, Cincinnati, Ohio, United States of America, 11 Division of Developmental Biology, Cincinnati Children's Hospital Medical Center, Cincinnati, Ohio, United States of America

* emily.miraldi@cchmc.org

**Data Availability Statement:** The maxATAC codebase is available from https://github.com/MiraldiLab/maxATAC, including basic usage

## Abstract

Transcription factors read the genome, fundamentally connecting DNA sequence to gene expression across diverse cell types. Determining how, where, and when TFs bind chromatin will advance our understanding of gene regulatory networks and cellular behavior. The 2017 ENCODE-DREAM *in vivo* Transcription-Factor Binding Site (**TFBS**) Prediction Challenge highlighted the value of chromatin accessibility data to TFBS prediction, establishing state-of-the-art methods for TFBS prediction from DNase-seq. However, the more recent Assay-for-Transposase-Accessible-Chromatin (ATAC)-seq has surpassed DNase-seq as the most widely-used chromatin accessibility profiling method. Furthermore, ATAC-seq is the only such technique available at single-cell resolution from standard commercial platforms. While ATAC-seq datasets grow exponentially, suboptimal motif scanning is unfortunately the most common method for TFBS prediction from ATAC-seq. To enable community access to state-of-the-art TFBS prediction from ATAC-seq, we (1) curated an extensive benchmark dataset (127 TFs) for ATAC-seq model training and (2) built "**maxATAC**", a suite of user-friendly, deep neural network models for genome-wide TFBS prediction from ATAC-seq in any cell type. With models available for 127 human TFs, maxATAC is the largest collection of high-performance TFBS prediction models for ATAC-seq. maxATAC performance extends to primary cells and single-cell ATAC-seq, enabling improved

(maxATAC installation, ATAC-seq data processing and TFBS prediction with the trained maxATAC models) and advanced (model training and benchmarking). OMNI-ATAC-seq data generated by this study is available as accession GSE197009 in the GEO Database. The maxATAC benchmark is organized (QC'd, processed, ready-to-go) for community use from https://doi.org/10.5281/zenodo.6761768. All relevant data are within the manuscript and its Supporting Information files.

**Funding:** This work was supported by the National Institute of Health (www.nih.gov) [U01AI150748 to ERM, SP, MTW, LCK; R01AI153442 to ERM, AB; R21AI156185 to ERM; R01HG010730, U01AI130830, R01NS099068, R01GM055479, P01AI150585 to MTW; NIH R01AI024717, R01AR073228; R01DK107502, R01AI148276, U19AI070235, U01HG011172, and P30AR070549 to LCK]; and Cincinnati Children's Research Foundation (www.cincinnatichildrens.org/research/cincinnati) [ARC Award 53632 to MTW, LCK; Center for Pediatric Genomics grants to ERM, AB, MTW, LCK, SP]. The funders had no role in study design, data collection and analysis, decision to publish, or preparation of the manuscript.

**Competing interests:** I have read the journal's policy and the authors of this manuscript have the following competing interests: AB is a co-founder of Datirium, LLC.

TFBS prediction *in vivo*. We demonstrate maxATAC's capabilities by identifying TFBS associated with allele-dependent chromatin accessibility at atopic dermatitis genetic risk loci.

## Author summary

Proteins called transcription factors interpret the genome, reading both DNA sequence and chromatin state, to orchestrate gene expression across the diversity of human cell types. In any given cell type, most chromatin is "inaccessible", and only those parts of the genetic code needed or likely to be needed soon are "accessible" for transcription factor binding to affect gene expression and cellular behavior. Hundreds of transcription factors are expressed in a given cell type and context (e.g., age, disease), and knowledge of their context-specific DNA binding sites is key to uncovering how transcription factors regulate cellular behaviors in health or disease. However, experimentally profiling the >1,600 human transcription factors across all cell types and contexts is infeasible. We built a suite of computational models "**maxATAC**" to predict transcription factor binding from a measurement of accessible chromatin, **ATAC-seq**. Importantly, ATAC-seq is feasible even at single-cell resolution. Thus, this data type, in combination with maxATAC, can be used to infer transcription factor binding sites in directly-relevant cell types isolated from physiological and disease settings, enabling insights into disease mechanisms, including how genetic variants and cellular context impact transcription factor binding, gene expression patterns and disease risk.

This is a *PLOS Computational Biology* Methods paper.

## Introduction

Most disease-associated genetic polymorphisms fall outside of protein-coding sequences [1]. Instead, they overlap significantly with enhancers, promoters and other locus-control regions [2]. Causal variants are thought to contribute to disease phenotypes by altering gene transcription in specific cell types [3,4]. Gene regulatory networks (**GRNs**) describe the control of gene expression by transcription factors (**TFs**) at genome-scale [5]. GRN reconstruction for human cell types will thus be crucial to identifying how noncoding genetic variants contribute to complex phenotypes through altered TF binding, chromatin looping and other gene regulatory mechanisms.

The Assay for Transposase Accessible Chromatin (**ATAC-seq**) opens new opportunities for GRN inference and genetics. Relative to other assays for chromatin accessibility (e.g., FAIRE-seq [6], DNase-seq [7]), ATAC-seq is an easy-to-use, popular technique that provides high-resolution chromatin accessibility with low sample input requirements [8]. Thanks to advances in single-cell (**sc**)ATAC-seq, it is now possible to resolve the chromatin accessibility profiles of individual cell types from heterogeneous tissues and limited clinical samples [9–11]. ATAC-seq is the only single-cell chromatin accessibility assay available via standard commercial platforms [12]. Whether from single cells or "bulk" populations, integration of TF-binding predictions from ATAC-seq improves GRN inference [13,14]. Although other experimental approaches more directly measure TF occupancy (e.g., ChIP-seq), they require substantial optimization, are costly in time and reagents, and are sometimes impossible due to a lack of quality antibodies. Indeed, hundreds of TFs are expressed in a given cell-type condition, but

profiling of >50 TFs has been accomplished for very few human cell types [15]. Furthermore, for some rare cell types and physiological conditions, limited sample material precludes direct measurement of TF occupancies.

Thus, the computational community collectively pioneered methods to predict TF binding sites (**TFBS**) from chromatin accessibility [16]. In 2017, the ENCODE-DREAM *in vivo* TFBS Prediction Challenge established two top-performing TFBS prediction algorithms [17,18] that vastly improved performance over popular motif scanning (median area under precision-recall .4 versus .1). Subsequent efforts (Leopard [19], DeepGRN [20], scFAN [21], and TAMC [22]) leveraged deep neural network modeling, yielding further advances in TFBS prediction from chromatin accessibility. Yet these top-performing TFBS methods are not used in popular ATAC-seq analysis pipelines [19–21,23]. Instead, TF motif scanning in accessible chromatin regions is the norm, and this less accurate method for TFBS prediction ultimately undermines biological inferences (TF enrichment, GRN and genetic analyses), experimental validation rates and overall scientific advances from ATAC-seq and scATAC-seq data.

Several factors limit community access to state-of-the-art TFBS prediction from ATAC-seq:

1. Current state-of-the-art models for TFBS prediction from chromatin accessibility were trained on DNase-seq data [17–20,23]. Although both DNase-seq and ATAC-seq provide high-resolution accessibility data, there are notable differences between the technologies [24,25]. Furthermore, the ENCODE-DREAM Challenge DNase-seq training data are single-end sequenced, while paired-end sequencing, which improves mappability, is the standard for modern ATAC-seq datasets [8,12]. Thus, it is risky to assume that the DNase-seq-trained models will perform well on ATAC-seq inputs.

2. The top-performing DNase-seq TFBS models exist for at most 32 TFs [17–20,23], while a recent deep learning method for scATAC-seq limited benchmarking of trans-cell type TFBS prediction to 17 TFs [21]. In contrast, TF motifs are available for at least 1200 of the ~1600 human TFs [26]. Thus, exceedingly few biologists are enticed to integrate the small number of state-of-the-art TFBS models into their ATAC-seq analysis pipelines.

3. While benchmark and training datasets have been developed for TFBS modeling from DNase-seq, such resources have not been curated, quality-controlled and organized for TFBS modeling from the more popular ATAC-seq assay.

4. Finally, development and maintenance of user-friendly software for state-of-the-art TFBS prediction is an on-going challenge.

Despite the promise of ATAC-seq for GRN inference and human genetics, TFBS prediction from ATAC-seq remains primitive, even as ATAC-seq data generation grows exponentially in both basic and clinical research.

To enable state-of-the-art TFBS prediction from ATAC-seq, we built "**maxATAC**", a collection of user-friendly deep neural network models for genome-scale TFBS prediction from ATAC-seq (**Fig 1**). maxATAC currently includes models for 127 human TFs, making it the largest collection of high-performance TFBS models available. This effort required extensive curation of existing ChIP-seq and ATAC-seq datasets as well as select data generation. Benchmarking led to methodological advances for TFBS prediction from ATAC-seq. As a result, our models perform well on both bulk and single-cell ATAC-seq, expanding maxATAC TFBS capabilities to scarce cell types *in vivo*. We use maxATAC to discover TFBS associated with allele-dependent chromatin accessibility at atopic dermatitis genetic risk variants, showcasing the potential for maxATAC to uncover molecular regulatory mechanisms in complex diseases.

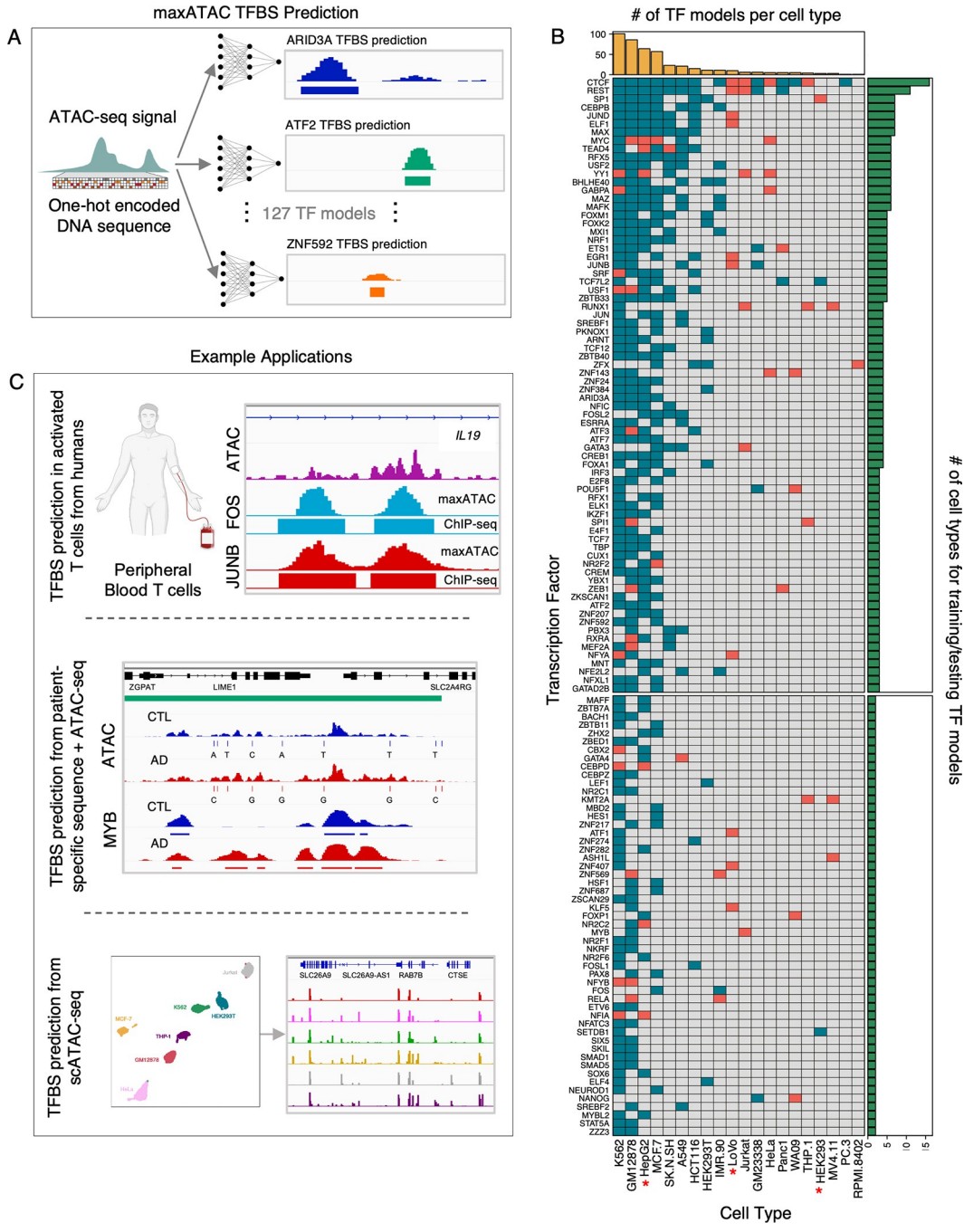

**Fig 1. Overview of maxATAC. (A)** maxATAC deep neural network models use DNA sequence and ATAC-seq signal to predict TFBS in new cell types. **(B)** The maxATAC training data per TF and cell type with OMNI-ATAC-seq (top: 74 "benchmarkable" TF models with ≥ 3 cell types available, bottom: 53 TF models with only 2 cell types for training). Teal boxes indicate ChIP-seq from ENCODE, while red boxes indicate data from GEO. Red stars denote cell types for which we generated OMNI-ATAC-seq. **(C)** Example applications of maxATAC TFBS prediction to primary cells, scATAC-seq and clinical studies combining DNA sequencing with ATAC-seq. Human image was created with BioRender.com.

## Results

### An unprecedented benchmarking resource for cross-cell type TFBS prediction from ATAC-seq

Development and community adoption of state-of-the-art methods for cross-cell type TFBS prediction from ATAC-seq were the objectives motivating our study. Construction of a new benchmark resource was a necessary first step. We needed a benchmark dataset that (1) enabled model training on ATAC-seq and (2) would yield a compendium of TF models large enough to inspire wide-scale community adoption of state-of-the-art methods. Specifically, these needs were not met by the popular DREAM-ENCODE *in vivo* TFBS Prediction Challenge benchmark, which is limited to DNase-seq and only 12 TFs (see **Introduction**).

Cross-cell type TFBS prediction greatly benefits from training in multiple cell types [18]. Thus, for benchmarking, we sought paired TF-binding (e.g., ChIP-seq) and ATAC-seq data in at least three cell types for benchmarking (at least 2 cell types for training and 1 to test generalizability in a new cell type, **Methods**). Limiting to human cell types, we identified >18,000 existing ChIP-seq and ATAC-seq experiments annotated by CistromeDB [27] and ENCODE [15]. Because experimental perturbations can impact TF binding and chromatin accessibility, it was critical that the TF ChIP-seq and ATAC-seq were derived from the same experimental conditions. We excluded data from any perturbed experimental condition (e.g., growth hormone or drug stimulation, genetic modifications, including ectopic TF expression, see **Methods**). To reduce technical variability in our ATAC-seq training data, we restricted to the OMNI-ATAC-seq protocol [28]. (Relative to the original "standard" ATAC protocol [8], OMNI-ATAC-seq has improved signal-to-noise ratio, better recovery of enhancers and reduced mitochondrial contamination.) For improved mappability, we required paired-end sequencing. We manually verified the annotations of each experiment, excluding 2,887 experiments that had experimental perturbation or were incorrectly annotated (**S1 Table** and **Methods**). Finally, we processed data, eliminating experiments that failed our quality-control metrics (**Methods**).

Data curation and quality-control of public data yielded an initial dataset of 361 TF-cell type combinations (110 TFs with at least 2 training examples across 17 cell types). Furthermore, we identified 3 cell lines with abundant TF ChIP-seq but no OMNI-ATAC-seq (HepG2, HEK293, LoVo). To expand the benchmark further, we generated OMNI-ATAC-seq for these cell lines, bringing our final maxATAC benchmark dataset to 438 unique TF-cell type pairs (**S1A Fig**).

In total, the benchmark spans 463 ChIP-seq and 55 ATAC-seq experiments across 20 cell types, facilitating benchmarking of 74 TF models ($\geq$3 cell types) and model construction for 127 TFs ($\geq$2 cell types, **Fig 1B**). The 127 maxATAC TFs span 35 TF families, with up to 26 TFs represented per family (**S1B Fig**). This unique, expansive resource enables rigorous benchmarking of cross-cell type TFBS prediction from ATAC-seq. The underlying quality-controlled, processed datasets are organized and available for community development of ATAC-seq-based TFBS prediction methods (see **Data Availability**).

### maxATAC architecture and "peak-centric, pan-cell" training approach

Deep convolutional neural networks (**CNNs**) provide top performance for many sequence-based prediction tasks, including prediction of TFBS [21,29–31]. They require no prior knowledge of TF motifs and instead learn complex patterns in input DNA sequence (nonlinear combinations of what often look like TF motifs) de novo. We thus chose deep CNNs to model TFBS from ATAC-seq and DNA sequence (**Fig 2** and **Methods**). Given the sparsity of our

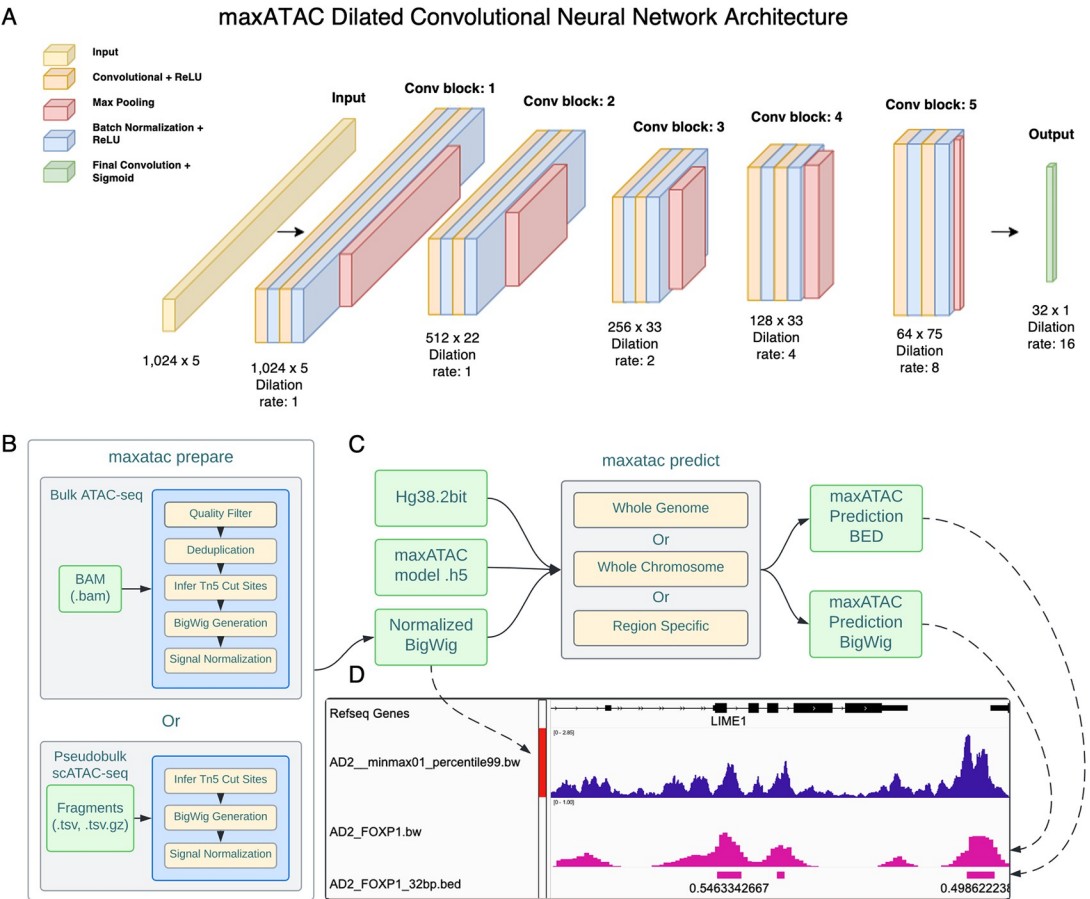

**Fig 2. maxATAC model architecture, inputs, and standard workflow. (A)** maxATAC inputs are a 1,024bp one-hot encoded DNA-sequence with ATAC-seq signal for the corresponding region, while maxATAC output is an array of 32 TFBS predictions at 32bp resolution, spanning the 1024bp input sequence interval. Inputs go through a total of 5 convolutional blocks. Each convolutional block consists of two layers, each composed of ReLU-activated, 1D convolutional operations and batch normalization. A max pooling layer is interspersed between the convolutional blocks to reduce the spatial dimensions of the input. The kernel width is fixed at 7 across all convolutional blocks. The model uses 15 filters in the first convolutional block, and the number of filters is increased by a factor of 1.5 for every subsequent block. The dilation rate of the convolutional filters increases from one, one, two, four, eight, to sixteen across blocks. Increasing the dilation rate increases the receptive field, so that spatially distant regions share information. In this network, the receptive field grows to +/-512bp, with information sharing proportional to spatial proximity. The final output is produced by a single sigmoid-activated convolutional layer. **(B)** Schematic overview of a standard maxATAC workflow. maxATAC takes as input a BAM file or scATAC-seq fragments TSV file that is processed to Tn5 cut sites, smoothed and converted to a read-depth-normalized ATAC-seq signal track (robustly min-max normalized between 0–1, see **Methods**). **(C)** The maxATAC predict function takes as input the genome reference DNA 2bit sequence file, a trained maxATAC model h5 file and the normalized ATAC-seq signal track to predict TFBS. **(D)** The outputs of maxATAC are a bigwig file of maxATAC TFBS scores, ranging 0–1, and a BED file of predicted TFBS, thresholded according to a user-selected confidence cutoff (e.g., precision, F1-score, see **Methods**).

training data (**Fig 1B**), we opted for single-task models (one CNN model per TF). We utilize dilated convolutional layers to capture spatially distant relationships across the input sequences in a multiscale manner [31,32]. This approach enables both high-resolution TFBS predictions (32bp) and information-sharing between proximal sequence and accessibility signals (+/-512bp).

Given our goal to construct TF models for as many TFs as possible, we developed training approaches to improve performance from minimal data (e.g., only 2–3 training cell types). We built a training strategy to enrich for true positive (**TP**) and challenging true-negative (**TN**)

examples of TFBS (**S2 Fig**). Challenging TN, for example, might arise when a TFBS in one cell type is not a TFBS in another cell type, due to potentially subtle differences in the chromatin environment. Because only ~1% of the chromatin is expected to be accessible or bound by TFs in a given cell type [33], randomly sampling genomic intervals for model training is inefficient (i.e., results in too few TFBS examples) [20]. Thus, for each TF model, we defined "regions of interest" as the union of accessible chromatin and TFBS (i.e., ATAC-seq and ChIP-seq peaks) for each training cell type. We found that increasing the representation of accessible chromatin and TFBS in training examples improved performance, relative to randomly selected genomic regions (**S2A Fig**). We refer to this practice as "peak-centric" training. Furthermore, we introduced "pan-cell" training, to increase the number of challenging TN examples. In pan-cell training, regions of interest (e.g., TFBS) in one cell type are equally likely to be selected from the other training cell type(s), which often do not share the TFBS. In this way, the training examples are enriched for challenging TN examples. Peak-centric, pan-cell training outperformed random sampling, with particularly strong gains for smaller training dataset sizes (**S2A Fig**).

## maxATAC models offer high-performance TFBS prediction from ATAC-seq at genome scale

For benchmarking, we compared maxATAC TFBS predictions to "gold-standard" TF ChIP-seq experiments in independent test cell types and chromosomes (**Methods**). By using all possible train-test cell type splits, we report a distribution of precision-recall statistics [34] for each of the 74 "benchmarkable" TFs (**Fig 3A and 3B** and **S2 Table**). Across these 74 TF models, median AUPR is .43, and median precision at 5% recall is .85.

The maxATAC models advance TFBS prediction from ATAC-seq. We first compared maxATAC model performance to the most popular method of TFBS prediction, TF motif scanning in ATAC-seq peaks [35–38] (implementation described in **S3 Fig** and **Methods**). maxATAC outperformed standard motif scanning for every TF (**Figs 3C and S4A**). Five of the maxATAC TF models (GATAD2B, NFXL1, ZBTB40, ZNF207, and ZNF592) have no characterized motif in the CIS-BP database [39], suggesting that prediction for these TFs might be especially challenging (motif-less models are highlighted with red dots in **Fig 3C**). Indeed, three of the models (NFXL1, ZNF207, ZNF592) had the 1st, 4th and 5th-lowest test AUPRs. However, two models had good test performance: ZBTB40 at $AUPR_{median}$ = .42 and GATAD2B at $AUPR_{median}$ = .31.

Some motif-based TFBS prediction methods (e.g., HINT-ATAC [25], TOBIAS [40], TAMC [22]) use ATAC-seq cleavage patterns to identify TF "footprints", or chromatin regions within ATAC-seq peaks that are protected from transposase cleavage due to TF binding. Here, we compared maxATAC performance to TOBIAS. maxATAC outperformed TOBIAS in terms of both AUPR (55 out of 55 TFs) and precision at 5% recall (53 out of 55 TFs) (**S4B and S4C Fig** and **S2 Table**). (Note we verified that TOBIAS outperformed simple TF motif scanning in terms of precision at 5% recall (44 out of 55 TFs), but not AUPR (**S4C and S4D Fig** and **S2 Table**).

For the vast majority of TFs (61 of 74), maxATAC also performed favorably to TFBS prediction using the average ChIP-seq signal for that TF across the training cell types (**Figs 3D and S4E**). Comparison to this simple model [41] ensured that the maxATAC models learned ATAC-seq and sequence features predictive of new TFBS in new cell types, not just TFBS in common with the training cell types.

Next, we compared maxATAC TFBS prediction to the state-of-the-art DNase-trained deep neural network model, Leopard [19], using ATAC-seq inputs across 8 test cell types for the 7

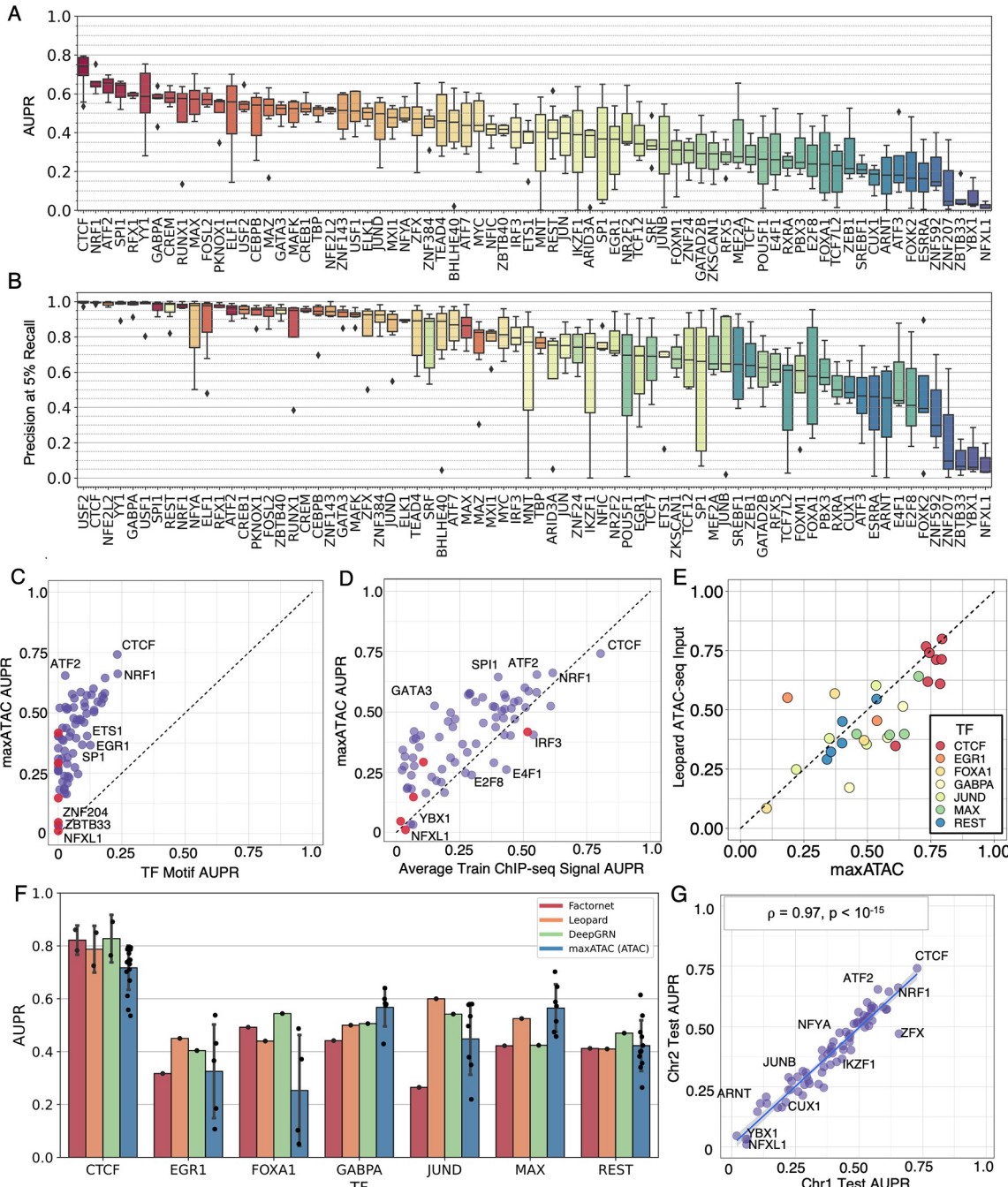

**Fig 3. The maxATAC models offer state-of-the-art TFBS prediction from ATAC-seq.** For every TF model, one cell type and two chromosomes (chr1, chr8) were held out during training to assess predictive (test) performance in a new cell type. Test **(A)** AUPR (median = 0.43) and **(B)** precision at 5% recall (median = 0.85). Boxplots display median (horizontal line), interquartile range (box), 3-quartile range (whiskers) and points outside the 3-quartile range (diamonds). maxATAC model performance is compared to **(C)** TF motif-scanning in ATAC-seq peaks and **(D)** TFBS prediction using the averaged ChIP-seq signal from the training cell types; each dot represents $AUPR_{MEDIAN}$ across train-test cell type splits. Red dots indicate TFs with no known motifs in CIS-BP. **(E)** Test AUPR of maxATAC models compared to Leopard (DNase-seq-based) model using ATAC-seq input and maxATAC ChIP-seq gold standards for 8 cell lines and 7 TFs. maxATAC outperforms Leopard for 20 out of 29 test performance comparisons. **(F)** Test AUPR of the maxATAC models on ATAC-seq compared to test AUPR reported by state-of-the-art deep learning models (Factornet, Leopard and DeepGRN) on DNase-seq. **(G)** Validation performance ($AUPR_{MEDIAN}$) on chr2 (training cell types) as a function of test performance ($AUPR_{MEDIAN}$) on chr1 (held-out test cell type) (n = 74; $\rho_{Pearson} = 0.97$, P $< 10^{-15}$).

TFs in common between Leopard (ENCODE-DREAM Challenge) and maxATAC benchmarks. Leopard performance was nearly on-par with maxATAC's; maxATAC outperformed Leopard in terms of AUPR (20 of 29 comparisons) and precision at 5% recall (17 of 29 comparisons) (**Figs 3E, S4G and S2 Table**). Finally, acknowledging important differences in test data (chromatin accessibility technology, test cell types, gold-standard ChIP-seq), we verified that performances reported by current state-of-the-art TFBS methods: Factornet [23], Leopard [19] and DeepGRN [20] on DNase-seq were roughly on-par with maxATAC models on ATAC-seq, for the 7 TFs in common between the ENCODE-DREAM Challenge and maxATAC benchmarks (**Fig 3F**).

Given these positive benchmarking results, we extended maxATAC model construction to all 127 TFs in our training dataset. Because 53 of the TFs had only two cell types available for training (i.e., no opportunity for both cross-cell type training *and* estimation of test performance in a new cell type), we explored the relationship between validation AUPR and test AUPR for the 74 benchmarkable TF models. Here, validation AUPR corresponds to performance on validation chromosome 2 in cell types used for training and validation, while test AUPR corresponds to performance on held-out test chromosome 1 for a cell type independent of training and validation cell types (**Methods**).

We observed a nearly one-to-one correspondence between validation and test performance (**Figs 3G and S4H**). Given this strongly predictive relationship between validation and test performance, the final maxATAC models were constructed using all available cell types for model training. For each TF model, we estimated confidences for maxATAC scores based on interpretable validation performance metrics (precision, recall, F1-score), so that users can choose confidence cutoffs suited to their research goals (**Methods**). For example, in the context of GRN inference with the *Inferelator* [13], initial, noisy TFBS predictions from ATAC-seq are subsequently refined by gene expression modeling, so a GRN modeler might equally prioritize precision and recall (i.e., use F1-score). In contrast, a researcher interested in experimental validation of a TFBS at a particular locus might prioritize high precision. Interpretable confidence cutoffs are a unique aspect of the maxATAC software package, which we further benchmark in primary cells (below).

## maxATAC model performance gains correlate with cell-type specificity

Our large training dataset affords new opportunities to explore potential factors driving differential performance across TF models. Test (**Fig 3A and 3B**) and validation (**S4I and S4J Fig**) performance varied dramatically across TFs, with $AUPR_{MEDIAN}$ ranging from .75 for CTCF to .01 for NFXL1 (**Fig 3A**). Given (1) the close correlation between validation and test performance and (2) that validation performance is available for all 127 TF models (versus 74 for test), we analyzed performance variation for both validation and test.

Model performance showed a modest dependence on the number of training cell types available (**S5A Fig**). While the interquartile range for models trained with 5 cell types was above the overall median AUPR performance (.43), the interquartile range for models trained with 2–4 cell types contained the overall median. We credit robust prediction in the small training data set regime (2–4 cell types) to our peak-centric, pan-cell model training strategy (**Methods and S2 Fig**).

While all maxATAC models outperformed motif-scanning (**Fig 3C**), maxATAC performance was comparable to averaged train ChIP-seq signal for several TFs (**Fig 3D**). Relative performance (maxATAC versus averaged train ChIP-seq) was not simply explained by the number of training cell types (ChIP-seq experiments) available, suggesting a role for TF-intrinsic factors (**S5B Fig**). We hypothesized that training ChIP-seq signal would perform well for

TFs whose binding patterns changed little across cell types, while maxATAC integration of context (ATAC-seq) with sequence would be especially critical for TFs whose binding patterns varied across cell types. We used Jaccard overlap [42] of TFBS between pairs of training cell types as a proxy for cell-type specificity (**S5C Fig**). There was a strong relationship between cell-type specificity and maxATAC performance gains relative to average ChIP-seq signal ($\rho_{\text{Pearson}}$ = .53, P<$10^{-5}$). For cell types with very high specificity (average Jaccard overlap < .05), AUPRs were, on average, >2-fold higher for maxATAC versus average ChIP prediction. For TFs with many shared binding sites across cell types (average Jaccard overlap >.25), maxATAC and average ChIP signal performed similarly. Thus, maxATAC modeling is especially important for TFs with context-specific binding sites and a good choice for TFs with many shared TFBS across cell types.

We used model interpretation techniques to determine context-specific DNA sequence patterns (TF motifs) learned by the maxATAC models. We highlight two examples: GATA3 and CREM, which exhibit high (GATA3) and moderate (CREM) cell-type-specificity (**S5C Fig**). To identify the TF motifs driving maxATAC TFBS prediction, we applied the methods Deep-LIFT [43] followed by TF-MoDISco [44] to the final maxATAC models (trained using all available cell types). These analyses uncovered cell-type-shared and cell-type-unique motifs, which correlated with test performance:

- **GATA3.** Across the four training cell types, we identified three distinct motifs that correspond with known GATA motifs: (1) monomer, (2) dimer with 3bp gap and (3) dimer with 4bp gap (**S6A Fig**). The 3bp-gapped dimer motif was not detected in A-549. Interestingly, test AUPR for A-549 (.43) was lower than AUPRs for the other cell types (.51 (SK-N-SH), .54 (MCF-7) and .56 (Jurkat)), suggesting that a model constructed using SK-N-SH, MCF-7 and Jurkat might have lower test performance on A-549 due to recognition of a motif (the 3bp-gapped dimer motif) that was not bound by GATA3 in A-549. This suggests the existence of cell-type specific co-factors or post-translational modifications for GATA3 that are not present in A-549 to mediate DNA binding to the 3bp-gapped dimer motif.

- **CREM**. Across the three training cell types, we identified three motifs: (1) the classic AP-1 7-base motif ("AP-1 short"), (2) the classic AP-1 8-base motif ("AP-1 long"), which has an extra base pair between half-sites, and (3) "AP-1 (methyl)", an AP-1 motif with an extra base pair between half sites and the "TG" replaced by a likely methylated "CG", as is frequently observed in *in vitro* methylated DNA binding data [45] (**S6B Fig**). Although the three motifs are identified across each of the cell types, their frequency varies substantially. The frequency of the AP-1 short and AP-1 long motifs is roughly halved in HepG2, relative to GM12878 and K562, suggesting that the maxATAC-learned TF motifs don't explain CREM binding in HepG2 as well as GM12878 and K562. In line with this reasoning, HepG2 test AUPR (.53) is the lowest, relative to GM12878 (.58) and K562 (.64).

Collectively, these results demonstrate that the maxATAC models are learning both canonical and noncanonical TF motifs and suggest (1) that cell-type-specific TF binding patterns not observed in training cell types, limit test performance in cell types with that pattern (GATA3 example), and (2) differences in the frequencies of TF motifs recognized in training versus test cell types might impact generalizability (CREM example).

## maxATAC performance extends to scATAC-seq and other chromatin accessibility techniques

Having constructed the largest suite of TFBS deep neural network models for ATAC-seq, we next tested whether these models, trained on bulk data from cell lines, could perform well in

new domains: single-cell ATAC-seq and primary cells. The maxATAC models were specifically designed to improve prediction of TFBS in cell types from *in vivo* contexts, where limited sample material or cell sorting strategies would preclude experimental TFBS measurement. Thus, we evaluated the maxATAC models on scATAC-seq.

Clustering of individual cells into cell types and subpopulations is a key first step in scATAC-seq analysis. Next, for each cluster, accessibility per cell is summed to create "pseudo-bulk" ATAC-seq signal for each inferred population of cells. From pseudobulk profiles, regions of chromatin accessibility are detected and annotated with TFBS predictions in standard scATAC-seq pipelines [37,38]. Pseudobulk scATAC-seq profiles are natural inputs for TFBS prediction with maxATAC.

For benchmarking, we took advantage of an experiment in which nuclei from 10 cell lines were combined for scATAC-seq [37]. Seven of the cell lines overlapped our maxATAC benchmarking cell types (**Fig 4A**), providing the opportunity to estimate test performance of maxATAC on scATAC-seq for 63 models (**Fig 4 and S3 Table**). In a side-by-side comparison using the same test cell types, maxATAC performed nearly as well on scATAC-seq as population-level ATAC-seq (**Fig 4B and 4C**). As expected, maxATAC outperformed popular TF motif scanning (**Fig 4D**).

To guide experimental design and application of maxATAC to scATAC-seq, we next examined the relative performance of maxATAC to TF motif scanning as a function of pseudobulk library size, using GM12878 as an independent test cell type (**Figs 4E, 4F and S7**, see **Methods**). Depending on the scATAC-seq experiment, the median number of fragments per cell typically ranges from ~5k-10k. Thus, we explored pseudobulk library sizes ranging from 100M to 100k fragments, which roughly correspond to 20-40k to 10–20 cells, respectively. We wanted to know whether there was a pseudobulk library size cutoff at which the simpler method (TF motif scanning) would be preferable to maxATAC. As expected, smaller pseudobulk library sizes (undersampling) diminished maxATAC performance. For example, at library size of 1M (~100–200 cells), median AUPR = .23 is nearly halved relative to the initial library size (100M fragments, ~20-40k cells, median AUPR = .43). Despite this, maxATAC outperformed TF motif scanning at every library size cutoff, even for the sparsest simulated pseudobulk library (corresponding to ~10–20 cells). While this was expected for the AUPR performance metric (because maxATAC is genome-scale and therefore has a recall performance advantage over TF motif scanning), these down-sampling experiments importantly demonstrated superior maxATAC performance for the precision at 5% recall metric. At the smallest library size (100k), TF motif scanning provided 0% recall for the majority of TFs (31 of 55, see per-TF performance breakdown **S7 Fig**). This analysis helps set expectations for TFBS prediction from both maxATAC and TF motif scanning as a function of pseudobulk library size. These results nominate maxATAC as the preferable method even for small library sizes.

Given the good performance of the maxATAC models on scATAC-seq, we evaluated maxATAC performance on additional chromatin accessibility datasets. For these analyses, we again focused on the extensively characterized lymphoblastoid cell line, GM12878, which enabled us to benchmark 60 maxATAC TF models on ATAC-seq generated from four different protocols and variable cell-input conditions (**Figs 5 and S8A-S8E**). In addition, we assessed DNase-seq. The ATAC-seq protocols included the original "standard" ATAC-seq protocol [8], OMNI ATAC-seq [28] (used for maxATAC model training, due to improved signal-to-noise, recovery of enhancers and reduction in mitochondrial contamination), Fast-ATAC [11] (designed for use in primary blood cells as it requires only a single lysis step), and the most recently developed: scATAC-seq (10x Genomics platform) [12]. We also evaluated TFBS prediction by motif scanning, to help distinguish whether protocol-dependent

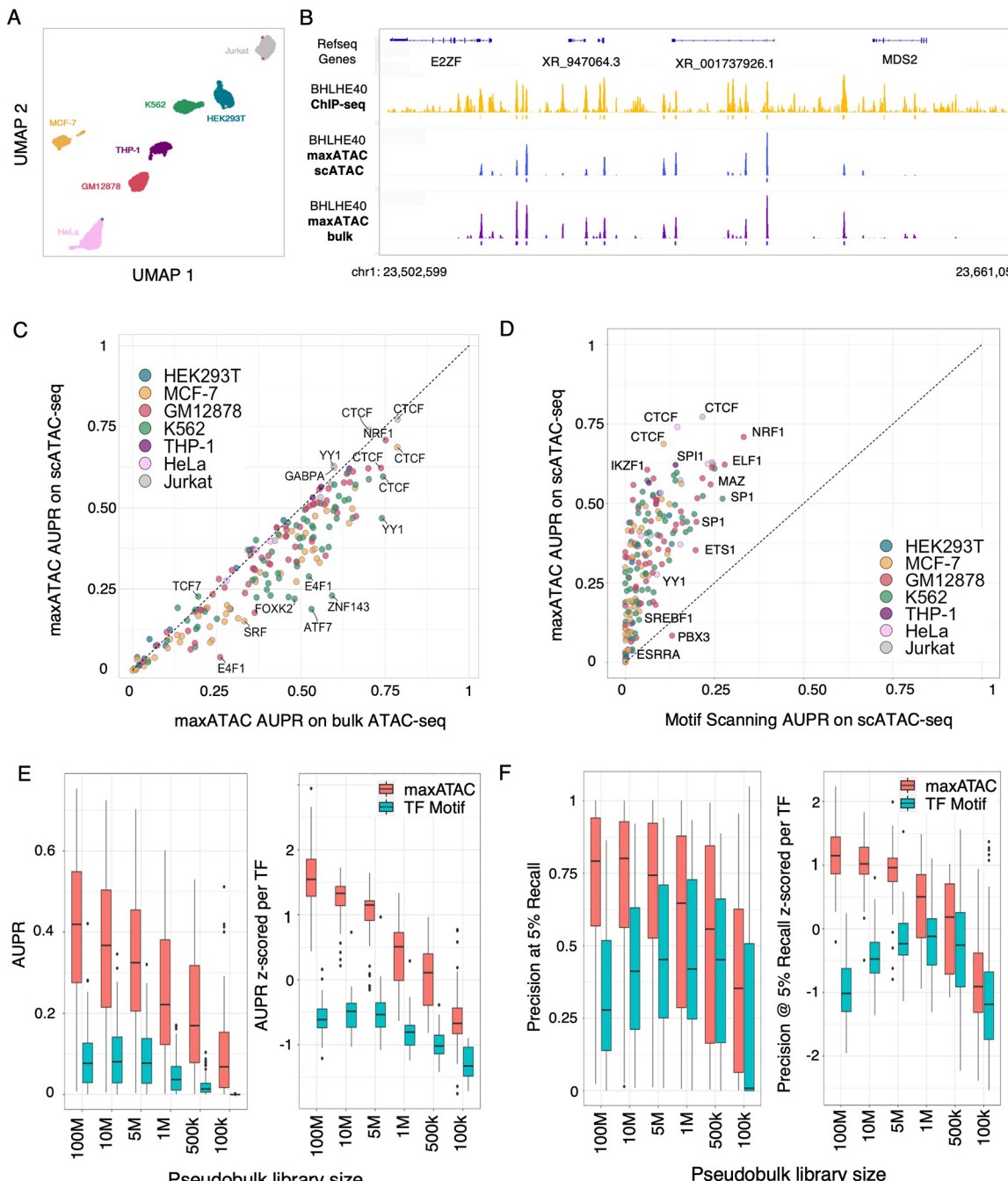

**Fig 4. maxATAC offers state-of-the-art TFBS prediction from scATAC-seq. (A)** UMAP of 10x scATAC-seq data from 7 cell types in a cell line-mixing experiment [37] that enabled test performance evaluation for 193 maxATAC models. **(B)** IGV tracks comparing BHLHE40 TFBS predicted by maxATAC in GM12878 from scATAC-seq (blue) or bulk ATAC-seq (purple), relative to BHLHE40 ChIP-seq ($-\log_{10}$(p-value) signal tracks in yellow) located at chr1:23,502,599–23,661,052 (158kb region). **(C)** Test AUPR for maxATAC in scATAC-seq relative to maxATAC performance on bulk ATAC-seq. **(D)** Test AUPR of maxATAC on scATAC-seq versus AUPR of TF motif scanning on scATAC-seq. Test **(E)** AUPR and **(F)** precision at 5% recall performances for maxATAC (red) and TF motif scanning (teal) as a function of down-sampled pseudobulk library sizes from scATAC-seq of GM12878 (n = 55 TFs with maxATAC models and TF motifs available). Given the range of performances per TF model, performances are also z-score normalized per TF, to better visualize model- and library-size-dependent trends.

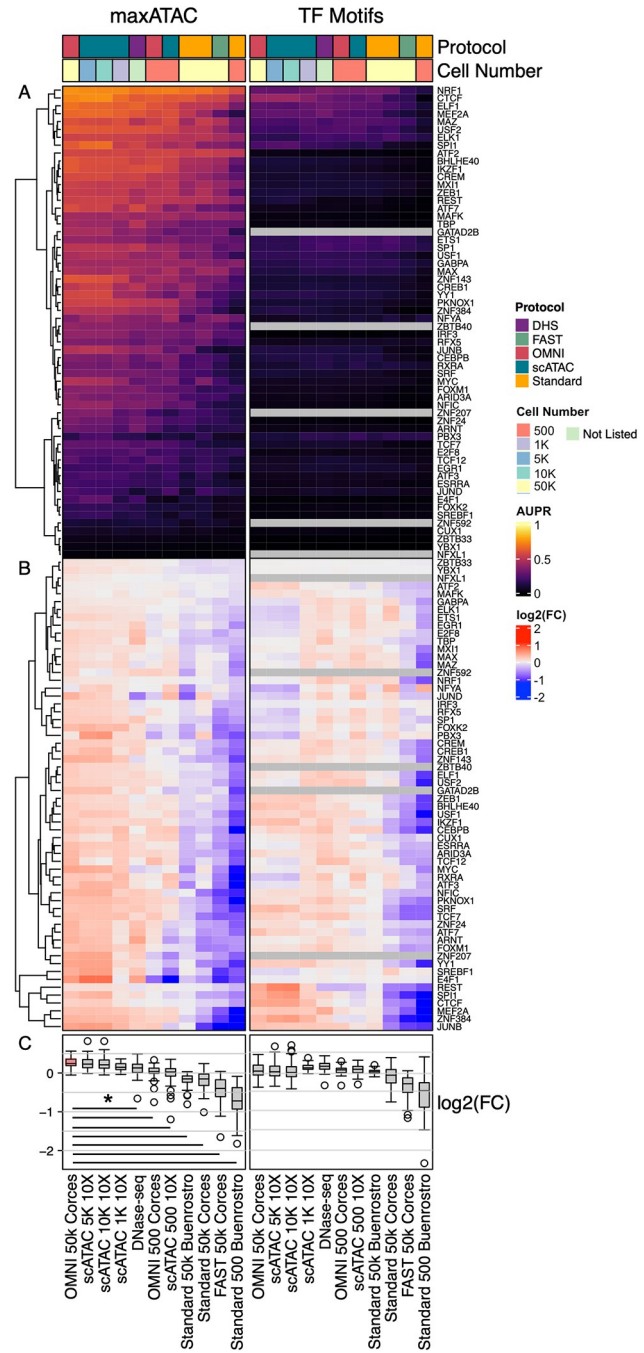

**Fig 5. Protocol and cell input numbers influence the performance of TFBS predictions. (A)** Test AUPR in cell line GM12878 for 60 TFs across chromatin-accessibility experimental designs (OMNI-, sc- and standard ATAC-seq or DNase-seq protocols, with variable input number of cells indicated). Grey squares indicate no predictions, due to lack of TF motif. TFs are hierarchically clustered based on maxATAC performance. **(B)** To visualize protocol-dependent trends for each method, AUPRs were normalized per TF (row-wise) as the $\log_2(AUPR:AUPR_{MEAN})$, independently for maxATAC and TF motif-scanning AUPRs. **(C)** Distribution of the $\log_2(AUPR:AUPR_{MEAN})$ per ATAC-seq sample. Given the maxATAC models were trained on OMNI-ATAC-seq ~50k cells, we compared each experiment to the reference "OMNI 50k Corces" sample (red boxplot). Black lines indicate protocol-dependent performance differences relative to the reference (Student's two-sided $t$-test, Bonferonni-corrected $P < 0.05$).

maxATAC performance was inherent to the quality of the assay for TFBS prediction (i.e., generalizable) or simply a limitation of maxATAC training on one assay type (OMNI-ATAC-seq).

For each experimental protocol, maxATAC dramatically outperformed motif-scanning (**Fig 5A**). To better visualize assay-dependent performance, we normalized AUPR performance per TF, independently for maxATAC and motif scanning predictions (**Fig 5B and 5C**). While maxATAC performance was indistinguishable for OMNI-ATAC-seq (50k cells) and scATAC-seq (1k-10k cells), performance was significantly worse for DNase-seq, OMNI-ATAC-seq (500 cells), scATAC-seq (500 cells), standard ATAC-seq (50k or 500 cells) and Fast-ATAC relative OMNI-ATAC (50k cells), the protocol used for maxATAC training data (two-sided Student's $t$-test, Bonferroni-corrected $P_{adj} < .05$). Library sizes were comparable (within 2-fold of the OMNI-ATAC-seq) for scATAC-seq (1k, 500 cells), standard ATAC-seq and DNase-seq (**S8C and S8D Fig**), suggesting performance differences among these experiments were not simply consequences of library size. Thus, maxATAC performance across all TF models is worse for standard ATAC-seq and DNase-seq relative to OMNI-ATAC-seq or scATAC-seq (1k-10k cells). In contrast, the performance of motif scanning was roughly comparable across standard ATAC-seq (50k cells), OMNI-ATAC-seq, DNase-seq and scATAC-seq. Collectively, these analyses suggest that technical differences between chromatin accessibility protocols undermine cross-protocol model prediction, but they also offer a simple solution: assay-specific model training for state-of-the-art prediction from that assay.

## maxATAC performance extends to primary cells

We next evaluated the performance of maxATAC on stimulated primary cells. In particular, we examined activation of naïve CD4+ T cells five hours following T-cell receptor (**TCR**) and CD28 stimulation, a timepoint at which standard ATAC-seq and ChIP-seq of TCR-dependent TFs (FOS, JUNB and MYC) had been collected [46] (**Fig 6A**). maxATAC predictions for all three TFs outperformed TF motif scanning in ATAC-seq peaks (**Fig 6B and 6D**). Even at low recall, maxATAC performance was on-par (JUNB) or better (MYC, FOS) than TF motif scanning and unparalleled at recall >10%. The TF motif-scanning precision-recall curves drop off suddenly at ~10% recall because motif scanning is not genome-scale (i.e., it is limited to ATAC-seq peaks). FOS and JUNB maxATAC predictions also outperformed TFBS prediction by average training ChIP-seq signal, while performance for MYC was slightly worse. This performance is especially promising, because the MYC and JUNB models perform worse on standard versus OMNI-ATAC-seq (1.3- and 3.5-fold AUPR decrease, respectively, **Fig 5A**). Thus, performance evaluation on standard ATAC-seq likely underestimates maxATAC performance on OMNI-ATAC-seq in new cell types.

We also investigated how the performance metrics associated with the FOS, JUNB and MYC maxATAC models extrapolated to this new test dataset. JUNB and MYC models had 4–5 benchmark cell types, and their associated test and validation performance metrics were good predictors of performance in the CD4+ T cells (**Fig 6E**). In contrast, the FOS model was constructed from only two cell types, and its performance, although superior to motif-scanning, was lower than estimated by validation performance. Aggregation of additional maxATAC training data will be the focus of future work.

We attribute maxATAC's generalizability to several key observations and methodological strategies. Good performance across ATAC-seq protocols required an extended genomic blacklist [47] and a robust normalization strategy [12]. In contrast to DNase-seq, ATAC-seq is variably contaminated with accessibility signal from mitochondrial chromosomes [24,48]. For example, we detected genomic regions of high chromatin accessibility in OMNI-ATAC-seq that were not present in scATAC-seq or DNase-seq (**S9 Fig**). Those regions shared high

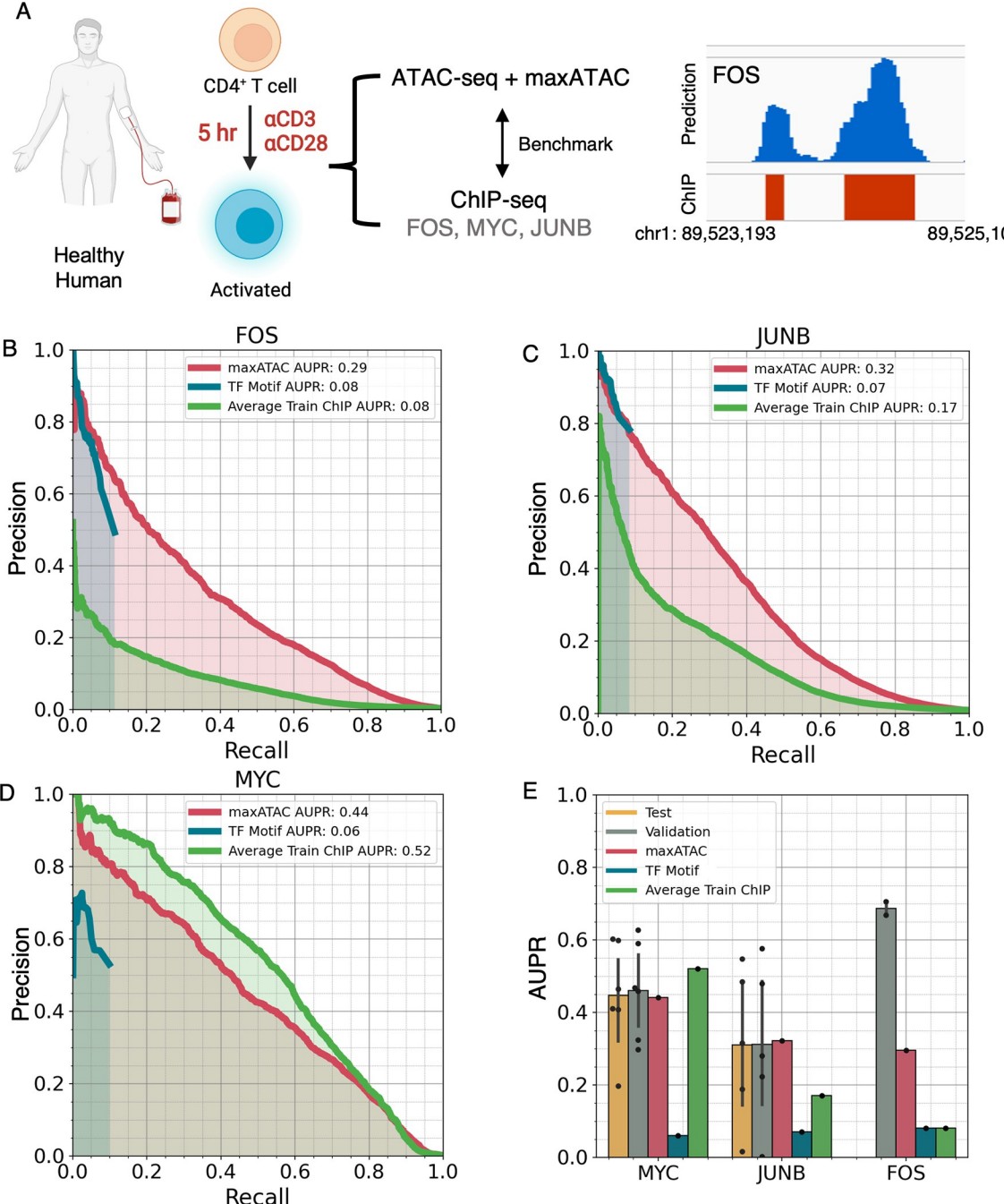

**Fig 6. maxATAC models perform well in primary human cells. (A)** Study design for maxATAC benchmarking in primary human CD4+ T cells. Precision-recall curves for TFBS predictions from maxATAC (red line), motif-scanning (blue) and averaged training ChIP-seq signal (green) for **(B)** FOS, **(C)** JUNB and **(D)** MYC, relative to experimentally measured TFBS (ChIP-seq). **(E)** Comparison of test performance in primary cells (red) to the estimates of test (green) and validation (yellow) performance (available with each trained maxATAC model) as well as TFBS prediction by TF motif scanning (blue) or average training ChIP-seq (green). Each point corresponds to a unique train-test cell type split. Error bars indicate standard deviation. Test performance minimally requires 3 cell types and therefore was available for MYC (n = 6 cell types) and JUNB (n = 5 cell types) but not FOS (n = 2 cell types). Human and cells image was created with BioRender.com.

sequence similarity with mitochondrial DNA, and, in general, we found less mitochondrial contamination in scATAC-seq than bulk. Thus, we expanded our blacklist to include chromatin regions with high mitochondrial sequence similarity. Despite the augmented blacklist, several of the transformed cell lines still had extreme, outlying ATAC-seq signal (**S10A–S10C Fig**). Some of these regions overlapped cancer-specific driver super enhancers (e.g., *TRIM37* locus in MCF-7). To be robust to these biological outlier regions, we normalized ATAC-seq data to the 99th-percentile highest signal (in contrast to the absolute max in standard min-max normalization; **Methods**); this strategy proved critical to maxATAC performance in scATAC-seq and primary cells (**S10D Fig**).

## maxATAC identifies allele-dependent TFBS at atopic dermatitis genetic risk variants

Identifying the cellular and molecular drivers of phenotypic diversity is a fundamental goal of basic and translational research. Complex traits are products of genetic and environmental factors. Chromatin accessibility is sensitive to environmental factors, like age [49] and microbiome [50], and therefore a critical complement to genetic profiling of patients across disease spectra, from cancer to autoimmune and obesity-related diseases. While previous work in deep neural network modeling focused on interpretation of genetic variants [30,31,51], maxATAC integrates both genetic (sequence) and environmental (ATAC-seq) signals and is therefore ideally suited to the elucidation of molecular drivers of diseases involving both genetic and environmental components.

We demonstrate the potential for maxATAC in the context of a complex disease. Atopic dermatitis (**AD**) is one of the most common skin disorders in children. Its etiology involves both genetic and environmental factors, with 29 independent AD risk loci known [52,53]. Here, we take advantage of an important genomics resource in AD: ATAC-seq and RNA-seq of activated CD4+ T cells, along with whole-genome sequencing, of AD patients and age-matched controls (**Fig 7A**) [54]. Previous analysis of this dataset identified several AD risk variants with allele-dependent chromatin accessibility in activated T cells [54].

Here, we use maxATAC to identify TFBS associated with allele-dependent chromatin accessibility at these AD risk loci (**S4 Table**). To avoid error associated with phasing of DNA and ATAC-seq signal (into maternal and paternal strands), we identified a pair of AD and age-matched controls ("AD2", "CTL2") where the AD patient was homozygous for the AD risk haplotype while the control was homozygous for the AD non-risk haplotype at two independent loci tagged by two variants: rs6062490 and rs1151624 (**Fig 7B** and **S4 Table**). For both variants, we considered the full haplotype block (linkage disequilibrium $R^2 > .8$), using donor-specific DNA sequence and accessibility to predict TFBS for 105 expressed TFs with maxATAC models available (**Figs 7C–7G** and **S11A**, see **Methods**).

The rs6062490 haplotype block is ~34.2kb and contains 30 noncoding SNPs interspersed between exons of two protein-coding genes: *RTEL1* and *TNFRSF6B* (**Fig 7D**). *TNFRSF6B* is an anti-apoptotic gene, and serum levels of this protein are increased in patients with atopic dermatitis [55,56]. The rs1151624 haplotype block is ~11.3kb and contains 19 noncoding SNPs cis to three genes: *ZGPAT*, *LIME1*, and *SLC2A4RG* (**Fig 7E**). *LIME1* is a transmembrane protein that controls effector T cell migration to sites of inflammation [57], while *SLC2A4RG* is a known eGene, (associated with an eQTL) in activated CD4+ T cells [58], regulatory T cells [59] and multiple other T cell subsets [60]. Thus, genes at both risk loci have ties to T cell biology, T cell eQTLs and AD.

For both loci, we ranked TFs based on the number of predicted differential TFBS regions between AD2 and CTL2 (**Figs 7F** and **S12**). Although TFBS were generally increased in the

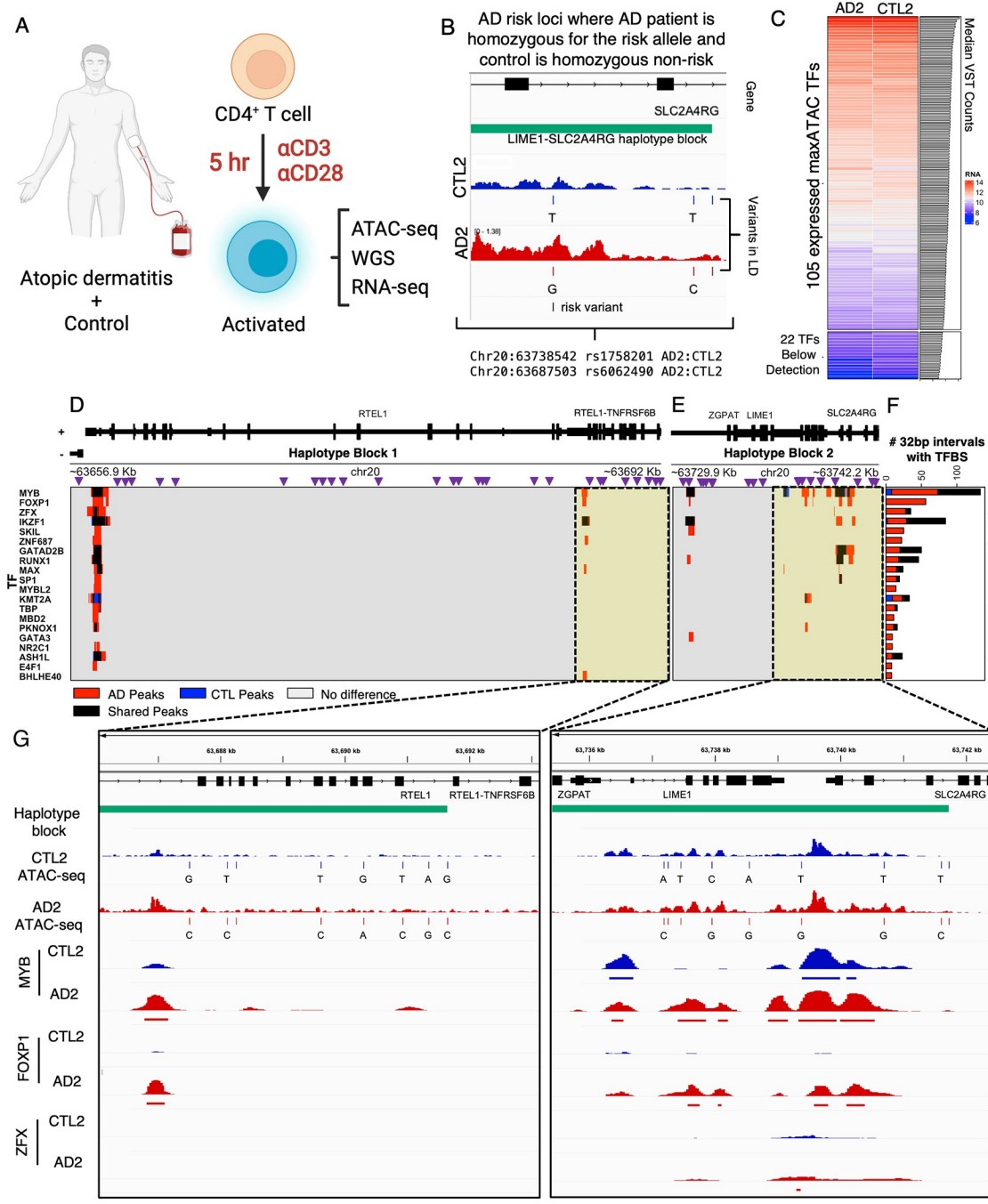

**Fig 7. maxATAC TFBS prediction at atopic dermatitis risk loci in patient-derived CD4⁺ T cells. (A)** In a previous study [54], peripheral CD4⁺ T cells were isolated from atopic dermatitis (**AD**) patients and age-matched controls (**CTL**) and TCR-stimulated prior to ATAC-seq, RNA-seq and whole genome sequencing (**WGS**) data generation. **(B)** We identified a pair of donors in which the AD patient (AD2) was homozygous for the risk allele and the age-matched control (CTL2) was homozygous for the non-risk allele at two independent loci: rs1758201 and rs6062490. **(C)** 105 of the 127 maxATAC TFs were nominally expressed for the donor pair, and these TFs were selected for TFBS prediction with maxATAC. We identified differential TFBS in the haplotype blocks containing **(D)** rs6062490 and **(E)** rs1758201. Purple triangles represent SNPs in linkage disequilibrium ($R^2 > .8$) with the AD risk alleles. In the heatmap below, red or blue intervals (32bp) indicate respective gain or loss of TFBS in the AD patient relative to control and black denotes intervals of shared TFBS. TFBS were determined using the cutoff that maximizes the predicted F1-score per TF model. **(F)** The 20 TFs with the greatest number of differential binding regions between AD2 and CTL2 are shown. **(G)** IGV screenshots showing regions (highlighted in yellow) of predicted differential TFBS in CTL2 (blue tracks) compared to AD2 (red tracks). The haplotype block is indicated in green. The top 4 signal tracks represent donor-specific ATAC-seq signal and genetic variants. The bottom 6 signal tracks represent predicted TFBS. Human and cells image was created with BioRender.com.

AD patient relative to the control in these loci (53 TFs), three TFs had a predicted decrease in binding sites in AD2 (KMT2A, LEF1, STAT5A), while 50 TFs had no TFBS predicted in these loci. MYB and FOXP1 showed the greatest differential binding, and their TFBS were predominantly increased in the AD patient. FOXP1 has been previously implicated in the maintenance of T cell quiescence and its expression is typically repressed in activated T cells [61]. MYB is a critical regulator of regulatory T cell differentiation and immune tolerance [62]. Thus, both TFs have known roles in T cell biology. We visualized regions of each haplotype block, putative cis-regulatory modules, where MYB and FOXP1 were predicted to bind with several other factors (**Figs 7G and S12**).

Although limited in statistical power, we examined the correlation between expression of genes at these loci in AD patients and age-matched controls (**S11B–S11G Fig**). We observed a trend for increased expression of *SL2A4RG* and decreased expression of *ZGPAT* in AD patients harboring the risk variant rs1151624 (trend held for all 4 homozygous-risk AD patients relative to their homozygous or heterozygous non-risk controls). We observed increased expression of all three genes *(RTEl1, RTEL1-TNFRSF6B, TNFRSF6B)* associated with rs6062490, when comparing the single pair of homozygous-risk AD to homozygous non-risk control, but we observed inconsistent trends from the two donor pairs in which the AD patients were homozygous risk and controls were heterozygous non-risk. These trends, in addition to support for *SLC2A4RG* as an eGene in CD4$^+$ T cells [58–60], nominate potential molecular mechanisms, specific TFs (MYB and FOXP1) that might alter the expression of genes important to T cell function in AD patients. Interestingly, for the regions shown in **Fig 7G**, neither FOXP1 nor MYB had motif occurrences in the differential ATAC-seq peaks, highlighting the unique predictive capabilities of maxATAC.

maxATAC analysis of this clinical genomics study provided "in silico ChIP-seq" for 105 TFs, in a setting where experimental measurement of TF occupancy for 105 factors was infeasible. Application of maxATAC to the growing number of genetic studies with population-level and single-cell ATAC-seq will improve the power of these studies to accurately predict TF mediators of allelic chromatin accessibility and gene expression. Furthermore, many genetic tools use overlap with TFBS to nominate potentially causal risk variants [63–65], where TFBS are predicted based on suboptimal TF motif scanning or ChIP-seq in a potentially suboptimal cell type or condition, due to lack of data in the *in vivo* disease context. Integration of maxATAC TFBS predictions into genetic analysis pipelines will be the focus of future work.

## Discussion

Genomic measurement and machine learning capabilities are advancing at an unprecedented pace. The possibilities for computational modeling to address fundamental questions in biology and human health have never been greater. Leveraging the fastest-growing chromatin-state measurement ATAC-seq, maxATAC seeks to make this moment's state-of-the-art in transcription-factor binding prediction readily achievable across the basic and biomedical research spectrum. Rigorously benchmarked across cell lines, primary cells, and single-cell ATAC-seq, maxATAC will improve TFBS predictions and knowledge gain from ATAC-seq studies. With maxATAC and its user-friendly codebase (available from https://github.com/MiraldiLab/maxATAC), genome-scale TFBS prediction can be accomplished for 127 human TFs using a single ATAC-seq or scATAC-seq experiment.

maxATAC is a unique resource, representing the largest collection of deep CNN TFBS prediction models for chromatin accessibility and a first-time benchmark of trans-cell type prediction on scATAC-seq. There are numerous deep CNN methods for ATAC-seq data [21,66–70], but these methods have different modeling objectives, including: prediction of ATAC-seq

signal from DNA sequence (e.g., AI-TAC [68], chromBPNet [66], deepMEL [67]), denoising of ATAC-seq and scATAC-seq signal (AtacWorks [70]), classification of enhancers, promoters and insulators based on ATAC-derived inputs (CoRE-ATAC [69]) and clustering of scATAC-seq data (scFAN [21]).

Many deep learning methods have been used to predict the impacts of genetics on epigenome signals, including TF binding (DeepSea [30], Basset [51], Basenji [31], Enformer [71], ChromBPNet [66]). However, DNA sequence is the *only* input to these models, and none make trans-cell type predictions (i.e., are capable of TFBS predictions outside of the cell types and conditions used for model training). In contrast, maxATAC predicts TFBS based on *both* DNA sequence and ATAC-seq signal, integrating both genetic and environmental effects to predict TFBS in new cell types and contexts (e.g., patient-specific scATAC-seq of a tumor biopsy). This is especially important for elucidation of complex disease mechanisms where environmental context contributes to disease etiology. Thus, maxATAC is a unique resource for TFBS prediction, with no comparable tools available–currently.

That said, we are confident that the training data we curated for maxATAC will spur rapid community advances in trans-cell type TFBS prediction from ATAC-seq. For this purpose, the maxATAC benchmark is organized (quality-controlled, processed, ready-to-go) for community download and development (https://doi.org/10.5281/zenodo.6761768).

We echo a top-performing group in the ENCODE-DREAM *in vivo* TFBS Prediction Challenge [17], acknowledging that state-of-the-art computational prediction of TFBS from ATAC-seq, at median AUPR = ~.4, is not yet accurate enough to replace TFBS measurement, when experimentally feasible. There are many avenues for improvement. For example, maxA-TAC models TFBS as binary, on-or-off events, despite the quantitative nature of the population-level TFBS measurements used for model training. Recent works [31,72] introduced new loss functions, specifically designed to quantitatively model chromatin state signal from NGS. Furthermore, BPnet's base-pair resolved architecture has already been tested on TF ChIP-seq [72] and appears well-poised to leverage highly-resolved ATAC-seq as model input, too.

TF binding *in vivo* is a multivariate process involving cooperation, competition, and co-binding among TFs. Yet each maxATAC TF model was trained independently, with predictions made for each TF one-at-a-time. In genomics, multi-task modeling of multiple TFs together is a common technique to enhance recovery of predictive sequence features, especially those involved in co-binding, relative to single TF models [73]. The sparsity of our training data (**Fig 1B**), combined with our goal to predict TFBS in new cell types, limited application of traditional multi-task learning approaches. Pre-training and transfer learning with large genomics resources could bridge the gap, providing richer, multi-task-like features for maxATAC [51], while we await experimental advances in massively parallel TF occupancy measurements. Relatedly, TFs mediate interactions between enhancers and promoters, yet, with maxATAC, each 1kb genomic interval is modeled independently. Chromatin-looping interactions could improve TFBS prediction, and, in the absence of context-specific experimental data, be inferred from existing looping data [74] or estimated from covariance of functional genomics assays [75] (including scATAC-seq [76]). 3D-chromatin interactions could be incorporated using graph neural networks [77,78] or with simple post-processing, like affinity propagation [79] of TFBS labels based on 3D contacts.

For some TFs, experimental-quality TFBS prediction will require more than ATAC-seq signal, especially for signal-activated TFs that bind pre-existing accessible chromatin. Thus, future directions for maxATAC will include addition of new data types, to implicate TFs based on transcriptional activity and methylation status of chromatin. Finally, the coverage of maxA-TAC models is still small relative to TF motif models (127 versus ~1200 for human TFs [26]). Thus, identification and generation of additional TFBS and ATAC-seq data is a top priority.

In summary, this work represents a significant resource, advancing community access to state-of-the-art TFBS prediction from popular ATAC-seq and scATAC-seq protocols.

## Materials and methods

### maxATAC training data

We curated training data for maxATAC from the CistromeDB [80] and ENCODE [15] databases (**Figs 1B and S1**), identifying cell types that had (1) a high number of human TF ChIP-seq experiments and (2) ATAC-seq data. We limited ATAC-seq training data to higher quality OMNI [28] and required paired-end sequencing, while, for ChIP-seq, we utilized both paired- and single-end sequencing. We required sequencing depth $\geq$ 20 million reads per biological replicate (sum of the spot numbers for SRR IDs per SRX ID). We manually verified the cell type, TF, and experimental source of each experiment. To ensure that ATAC-seq and ChIP-seq were derived from the same experimental conditions per cell type, we eliminated any ATAC-seq or ChIP-seq experiment in which the cell type was perturbed (including genetically modified, transfected with exogenous vectors, or treated with vehicle controls, environmental perturbations, metabolic manipulations, or differentiation protocols).

### ChIP-seq

When available, we utilized processed ENCODE ChIP-seq experiments over other publicly available data. For TF and cell-type pairs with multiple ChIP-seq experiments available, we chose the most recent experiment with the greatest number of reproducible TFBS (peaks) detected by IDR analysis [81]. If available, we selected conservative over optimal IDR peak sets. We excluded any experiment with a red flag (i.e., a"critical issue" was identified by ENCODE) as well as experiments with fewer than 500 TFBS detected. This resulted in 371 TF-cell type combinations from ENCODE.

The remainder of our ChIP-seq training data required processing and additional quality control. Using snakemake [82], we followed ENCODE3 [83] standards for ChIP-seq read alignment, read filtering, and peak calling; this workflow is available from the maxATAC codebase. In brief, each biological replicate was summarized according to SRX ID for a total of 316 experiments. Fastq files were downloaded (*SRAtools fasterq-dump* v. 2.10.8) with technical replicates concatenated per SRX ID. Fastq files were assessed for adapter contamination and read quality statistics (*FastQC* v. 0.11.9). Samples flagged for high levels of N sequences were removed. *TrimGalore!* (v. 0.6.7) was used to remove adapter contamination and trim the low-quality bases at the 3' end of the sequencing read with the settings (-q 20). Samples with (1) < 15 million reads after filtering or (2) average read length < 20bp after trimming were excluded from the analysis. Reads were aligned to the hg38 reference genome using *bowtie2* (v. 2.4.4) [84] (—very-sensitive—maxins 2000). The aligned reads were quality filtered with *samtools* (v. 1.9) [85] (-F 1804 -q 30) and PCR duplicates were removed with *samtools markdup* (-r -s). Prior to peak-calling, we also excluded reads mapping to blacklist regions compiled from ENCODE data [47] as well as centromeres, telomeres, and annotated gaps. MACS2 (v 2.2.7.1) [86] was used to call peaks on the filtered BAM file with parameters (—nomodel—extsize 147). For cell types and TF combinations with multiple biological replicates, we provided all filtered BAM files during peak calling. Peaks meeting a FDR = 5% cutoff were retained as TFBS for benchmarking. TF-cell type conditions with fewer than 500 TFBS detected were excluded.

Further QC of the "non-ENCODE" ChIP-seq involved TF motif analysis and biological replicate Pearson correlation of > .6 (when available). We used HOMER (v. 4.11) [36] with the CIS-BP (v. 2.0) [39] database to test for enrichment of expected motifs in ChIP-seq peaks.

CIS-BP provides multiple motifs per TF, so we selected the most highly enriched motif per TF and then ranked motif enrichment scores at the TF level. We excluded ChIP-seq experiments in which the ChIP'd TF was not ranked among the top-10 enriched TFs. From 316 experiments, we derived 72 TF-cell type pairs. In total, our 127 TF models are derived from 443 unique TF-cell type combinations across 20 cell types (**Figs 1B** and **S1A**).

## ATAC-seq

We combined public with in-house OMNI-ATAC-seq for our 20 benchmark cell lines. In-house, we ordered HepG2, LoVo and HEK293 cells from ATCC and targeted a median sequencing depth of 20 million reads for each cell type. OMNI-ATAC-seq was later released [83] for one of these cell types (HepG2), so we combined biological replicates for this cell type. To evaluate strategies for alignment, signal normalization and smoothing, we processed ENCODE along with in-house ATAC-seq. ENCODE ATAC-seq were downloaded, converted to FASTQ (*SRAtools fasterq-dump*) and then subsampled to a depth of 30 million reads per biological replicate, to limit compute time for alignment. For both ENCODE and in-house data, we evaluated sequencing quality with *FastQC*. Sequencing adapters and bases with a PHRED score < 30 were trimmed with the package *Trim Galore*! [87] using the parameters (-q 30 -paired). We excluded ATAC-seq experiments with < 20 million reads.

*Alignment*. We investigated several alignment strategies, using GRCh38 as the reference genome. We tested the performance of STAR (v. 2.7.0a) [88], bowtie2 (v. 2.4.4) [84], and bwa-mem (v. 0.7.17) aligners on TFBS predictions. Two STAR alignment strategies were tested, one with default parameters (—alignIntronMax 1—alignMatesGapMax 2000—alignEndsType EndToEnd) and the second with parameters from the TOBIAS [40] TF footprinting package (—alignIntronMax 1—alignMatesGapMax 2000—alignEndsType EndToEnd—outMultimapperOrder Random—outFilterMultimapNmax 999—outSAMmultNmax 1—outFilterMismatchNoverLmax 0.1—outFilterMatchNmin 20—alignSJDBoverhangMin 999—alignEndsProtrude 10 ConcordantPair). For STAR, a MAPQ score of 255 indicates properly paired reads with a single match and a samflag of 3 indicates properly paired and oriented reads. Thus, we filtered for STAR-aligned reads with a MAPQ score of 255 and samflag of 3 (*samtools view* -f 3 -b -q 255). For bowtie2, we used parameters (-p 8—very-sensitive—maxins 2000), while we used default parameters for bwa-mem, applying post-alignment filters for reads with MAPQ score of $\geq$ 30 and samflag 3. For all alignments, we removed duplicates (i.e., potential PCR artifacts) with *samtools rmdup* and *samtools fixmate* with parameter (-*n*), and further filtered for reads mapping to autosomal chromosomes. In contrast to normalization strategies, the maxATAC models performed robustly across the alignment methods tested (**S10D Fig**). We chose bowtie2 alignment for subsequent analyses.

*Inference of ATAC-seq Tn5 sites and smoothing*. The Tn5 transposase dimer inserts sequencing adapters with a strand-specific bias that results in a 9bp sequencing extension [8,24,89], therefore, reads are shifted +4 on the (+) strand or -5 on the (-) strand so that the corresponding read ends are centered at the Tn5 cut site. We first converted the filtered BAM files to bed intervals using *BEDtools bamtobed* [90] and awk (*awk 'BEGIN {OFS = "\t"}; {if ($6 = = "+") print $1, $2 + 4, $2 + 5, $4, $5, $6; else print $1, $3–5, $3–4, $4, $5, $6}'*). In contrast to other pipelines [37,38], we retain both cut sites per fragment, to maximize coverage and ultimately prediction from, e.g., scATAC-seq of rarer cell types *in vivo*. We used a 1bp window around the inferred cut sites to generate a high-resolution cut site signal. Given our goal of applying maxATAC to scATAC-seq in addition to ATAC-seq at typical sequencing depths (~20 million reads), we smoothed the sparse Tn5 cut site signal to overcome noise due to under sampling. We found that extension of Tn5 insertion sites by +/- 20bp (*BEDtools slop*) performed well (on

par with +/- 5 or 10bp and better than single-bp resolution); the resulting 40bp window also corresponds to the ~38bp wide Tn5 transposase dimer [89].

*Extended blacklist.* Initial testing of maxATAC models on scATAC-seq in GM12878 highlighted the need for an extended blacklist [47]. We discovered regions of extreme OMNI-ATAC-seq signal that were not present in DNase-seq or scATAC-seq data and therefore likely indicative of platform-specific technical artifacts. For example, regardless of alignment method, high signal regions were detected in OMNI-ATAC-seq but not DNase-seq or scATAC-seq of the same cell type (GM12878); these corresponded to mitochondrial chromosome duplication events and regions of low mappability (**S9 and S10 Figs**). Thus, our final blacklist included (i) blacklisted regions from ENCODE data [47], (ii) centromeres, telomeres, and annotated gaps available from UCSC table browser [91] for hg38, (iii) regions ≥1kb with ≥ 90% sequence identity to chrM [92], and (iv) regions with low mappability on chr21 (**S5 Table**). Inferred Tn5 cut sites within blacklisted regions were removed with *bedtools intersect*.

*ATAC-seq normalization and signal tracks.* For comparison of ATAC-seq signal tracks and combination of biological replicates, we scaled ATAC-seq signal per replicate to 20 million mapped reads (**RP20M**), a process involving signal conversion to BEDgraph interval coverage tracks (*bedtools genomecov*) using the scale factor, derived from **Eq 1** and **Eq 2**, and parameters (-bg -scale *scale factor*). The scale factor is multiplied by the count at each position to yield RP20M normalization:

$$RP20M\ scaled\ reads = \frac{count}{sequencing\ depth} \times 20,000,000 \qquad (1)$$

$$scale\ factor = \frac{1}{sequencing\ depth} \times 20,000,000 \qquad (2)$$

For cell types with multiple replicates, we average the RP20M values across all available samples, using (pyBigWig; v. 0.3.18) to generate bigwig files.

For maxATAC input, we initially applied standard min-max normalization to the RP20M ATAC-seq signal tracks, scaling bp signal by the min and max across RP20M tracks:

$$minmax_{P\%}(signal) = \frac{signal - min}{max_{P\%} - min}, \qquad (3)$$

where min corresponds to the minimum RP20M signal and $max_{P\%}$ corresponds to the pth-percentile (highest) RP20M signal. Standard min-max normalization uses the absolute maximum or "$max_{100\%}$". However, regions of extreme ATAC-seq signal persisted in a cell-type-specific manner, despite an ATAC-specific blacklist (**S10 Fig**). Although often biological in nature (e.g., *TRIM37* locus in MCF-7 [93], **S10C Fig**), these outlying signals interfered with min-max normalization of ATAC-seq and cross-platform performance on scATAC-seq (**S10D Fig**). To improve robustness, we replaced the absolute max (p = 100%) in **Eq 3** with the 95th-percentile and 99th-percentile signals (p = 95% or 99%). These strategies enabled high-quality prediction in initial tests of maxATAC on scATAC-seq in GM12878, while a standard max-min strategy did not (**S10D Fig**). This normalization strategy was then independently tested on another scATAC-seq dataset [37] (**Fig 4**).

*Peak Calling.* We called peaks with MACS2 [86] to identify "regions of interest" for training (see **Model Training**) and TFBS prediction with PWMs (a key comparator). The Tn5 cut sites (per biological replicate, when available) served as input to MACS2. Our parameter settings (-f BED -shift = 0 -ext = 40 -keep-dup = all) center the signal over the Tn5 insertion, smooth by

extension +/-20bp and ensure that each inferred Tn5 binding site contributes to the peak call. (We keep all duplicate Tn5 cut sites because PCR duplicates were removed in previous steps.) We retained ATAC-seq peaks per cell type if they met an FDR = 5% cutoff.

### DNase-seq

DNase-seq for GM12878 was downloaded from ENCODE experiment ENCSR000EMT. Alignment files were downloaded for both biological replicates mapped to the hg38 genome. The BAM files were deduplicated with Picard tools MarkDuplicates then sorted and filtered with samtools (-F 512 -q 10). DNase cut sites were inferred by centering a 40bp window on the 5' end of the read interval. Coverage tracks were generated as described in *ATAC-seq normalization and signal tracks*. MACS2 peaks were called using the cut sites for both biological replicates as described in *ATAC-seq Peak Calling*.

### Model architecture

The maxATAC models are deep dilated convolutional neural networks that predict transcription factor binding sites as a function of ATAC-seq signal and DNA sequence (**Fig 2**). Model inputs are 1,024bp x 5, with four dimensions corresponding to one-hot encoded DNA sequence and the fifth corresponding ATAC-seq signal (processed as described above). The output of each TF model is a 32 x 1 array of TFBS predictions, resolved to 32bp. The CNN is composed of five convolutional blocks, each consisting of two repeating double-layers (ReLU-activated 1D convolutional operations) followed by batch normalization. The max-pooling layer is interspersed between convolutional blocks to reduce the spatial dimensions of the input. The kernel widths are fixed at 7 for all convolutional blocks, while the number of filters grows by a factor of 1.5 per block, from 15 (first block) to 75 (last block). The final output layer uses sigmoid activation for binary prediction of TFBS. The dilation rate of the convolutional filters increases from one, one, two, four, eight, and sixteen across blocks. As a result, the receptive field gradually expands to +/-512bp in the ultimate hidden layer. Thus, information is shared across the length of the 1024bp input, in a distance-dependent manner (i.e., proportional to spatial proximity of the regions), while the resolution of TFBS predictions is preserved at 32bp.

### Model training

**Train, validation, and test sets.** The goal of maxATAC is TFBS prediction from ATAC-seq in new cell types (i.e., "trans-cell type TFBS prediction"). Thus, for a TF with TF ChIP-seq and ATAC-seq available from $N \geq 3$ cell types, N-1 cell types were used for training and validation, while the Nth was reserved for the test set. In addition, to avoid overfitting DNA sequence, the autosomal chromosomes were split into independent training (chromosomes 3, 4, 5, 6, 7, 9, 10, 11, 13, 14, 15, 16, 17,18, and 20), validation (chr2 and chr19) and test (chr1 and 8) sets. Given the 32bp resolution of maxATAC predictions, positive examples of TFBS resulted if the 32bp region had >50% overlap with the set of ChIP-seq TFBS for a given cell type (detailed above).

**Training routines.** The maxATAC models were trained through optimization of the cross-entropy loss function via stochastic gradient descent using the ADAM optimizer [94], with an initial learning rate of 0.001, for 100 epochs (batch size of 1000 and 100 batches per epoch) and Glorot initialization of the weights [95]. Given the breadth of TFs in our benchmark, the diversity of their binding mechanisms and variable amounts of training data available per model, we used maximum validation dice coefficient to select model parameters (an epoch) per TF model.

As described in **Results**, we developed a sampling strategy for model training, to enrich true positive (**TP**) and challenging true-negative (**TN**) examples of TFBS (**S2 Fig**). Because only ~1% of the chromatin is expected to be accessible in a given cell type, for each TF model, we defined "regions of interest" as the union of accessible chromatin and TFBS (for that TF) for each training cell type. Furthermore, to increase the number of challenging TN examples, we introduced "pan-cell" training, so that regions of interest for one cell type were equally likely to be selected from the other training cell type(s). Using a small subset of the benchmark data (11 TFs, limiting training to A549, HepG2, IMR-90, K562, MCF-7, and SK-N-SH cell lines, and test to GM12878), 100% peak-centric, pan-cell training outperformed commonly used random sampling from the genome and mixed random and peak-centric strategies (e.g., 50–50, **S2A Fig**). Thus, 100% peak-centric, pan-cell training was used for the final maxATAC models.

Following previous work [96], we doubled the number of training examples by including sequence and signal from the reverse-complement in addition to the forward strand. Although a part of our final maxATAC training routine, this strategy increased training time but did not robustly improve test performance. For this reason, it is not the default for model training in the maxATAC codebase.

## Performance evaluation

**Precision-recall analysis.** Genome-wide, the number of true positive (**TP**) TFBS are scarce relative to true negative (**TN**), unbound regions, so we used precision-recall statistics rather than receiver-operator characteristic (**ROC**) [34]. ROC weights TP and TN equally, and therefore is a less informative metric of performance in unbalanced classification problems, like TFBS prediction. We report area under precision-recall (**AUPR**) and precision at 5% recall.

We ranked maxATAC TFBS predictions for precision-recall analysis. maxATAC output is a score, ranging from 0 and 1, indicating the probability of a TFBS in each 32bp genomic interval. Each unique score is a unique rank. We benchmark our predictions by binning the signal from the validation or test chromosome(s) of interest using pyBigWig and report the max value per bin of length 200bp. (200bp was selected for comparison with the DREAM-EN-CODE TFBS Prediction Challenge.) The ranking of our predictions is based on the maximum score in the 200bp signal region.

Blacklisted regions are excluded from precision-recall analysis. The ChIP-seq gold standard is binned into 200bp intervals, and a bin is labeled positive if any of the bin overlaps a ChIP-seq peak. For every rank, we calculate precision and recall relative to the ChIP-seq gold standard. We calculate the precision as the number of predictions that were found in the gold standard at each rank (**Eq 4**). We calculate recall as the percent of the gold standard that was recovered at each rank (**Eq 5**). Random precision is calculated as the number of bins overlapping the gold standard divided by the total number of bins evaluated (**Eq 6**). AUPR calculations were implemented using the python package *sklearn* [97].

$$precision = \frac{\text{of bins with TF binding predictions overlapping ChIP} - seq\ gold\ standard}{\text{of bins with TF binding predictions}} \quad (4)$$

$$recall = \frac{\text{of bins with TF binding predictions overlapping ChIP} - seq\ gold\ standard}{\text{of bins overlapping ChIP} - seq\ gold\ standard} \quad (5)$$

$$random\ precision = \frac{\text{of bins overlapping ChIP} - seq\ gold\ standard}{\text{of bins across the chromosome}} \quad (6)$$

For $\log_2$(fold-change) comparisons between AUPR or precision of different methods, we added a pseudocount of .1, to moderate high fold-changes due to small numbers from relatively poorer quality models (e.g., AUPRs < .1).

## Comparison to other TFBS methods

*TFBS prediction with PWM models.* We obtained DNA sequences for ATAC-seq (or scATAC-seq) peaks identified per cell type (*bedtools getfasta*). We used the motif-matching algorithm MOODS [98] together with the TF PWM database CISBP v2 [26] to identify motif occurrences with a $P < 10^{-5}$. For TFs with multiple PWM, we used all for our analysis, and, when multiple motif matches occurred within the same genomic region, we removed exact coordinate duplicates, but left overlapping motif matches. To rank TFBS predictions (e.g., at 200bp resolution), we binned the genome into set-width, non-overlapping bins using *bedtools makewindows* (**S3A Fig**). For each bin, we counted the number of TF motif matches that overlapped the bin by at least 1bp using *bedtools intersect* (**S3B and S3C Fig**). The gold standard bins were defined by intersecting ChIP-seq peaks with the genomic windows (**S3D Fig**). The number of motif matches per bin were used to rank our predictions for precision-recall analysis (**S3E and S3F Fig**).

*Comparison to TOBIAS footprinting.* Following the TOBIAS footprinting pipeline [40], we aligned GM12878 OMNI-ATAC-seq data to the hg38 genome. The TOBIAS protocol recommends a library of motifs in which each TF is represented by a single motif (i.e., one motif per TF). We used the JASPAR vertebrate core motif set [40], and, for TFs with more than one motif, we report performance of the motif with the maximum AUPR.

*TFBS prediction with Average Training ChIP-seq Signal.* We used average ChIP-seq signal from the training cell types to predict TFBS in a held-out test cell line. Specifically, for each TF, we averaged the arcsinh of the ChIP-seq signal p-value across training cell types at each genomic position [99], using the averaged signal to rank TFBS for precision-recall analysis.

*Leopard.* We compared maxATAC model performance to the state-of-the-art method Leopard. Leopard models were trained on DNase-seq and ChIP-seq data aligned to the hg19 reference genome. In addition to one-hot encoded DNA and DNase-seq signal, Leopard also takes as input a signal track representing the average accessibility signal across multiple cell types. To compare Leopard prediction to maxATAC, we aligned ATAC-seq in our benchmark to the hg19 genome. ATAC-seq reads were processed to bigwig files representing filtered alignments with deeptools [100] bamCoverage (-bs 20—minMappingQuality 30). ATAC-seq samples were sampled for quantile normalization with the Leopard script subsample_for_qn.py (-rg grch37) and normalized to the liver sample with quantile_normalize_bigwig.py (-rg grch37), using both biological replicates as inputs. Leopard was used to predict TF binding for chromosome 1 using the complete mode (-tf {TF} -te {test_cell_line} -chr chr1 -m complete). The output of Leopard is a numpy array of predictions at base-pair resolution that were parsed to bedgraph and bigwig files, for precision-recall analysis using the maxATAC codebase. A modified version of Leopard and code for parsing outputs are available at https://github.com/tacazares/Leopard.

*Comparison to current state-of-the-art deep-learning models for trans-cell type TFBS prediction.* For the 8 TFs in common between the ENCODE-DREAM benchmark (DNase-seq train/test data, mapped to hg19) and the maxATAC benchmark (**Fig 1B,** ATAC-seq train/test data, mapped to hg38), we compared test AUPRs (at 200bp resolution) for maxATAC on the maxATAC benchmark to test AUPRs (also 200bp resolution) of DeepGRN, FactorNet and Leopard on the ENCODE-DREAM benchmark, reported in their respective publications [19,20,23].

## Prediction with the maxATAC models

The final maxATAC models were constructed using all benchmark cell types available for a given TF (**Fig 1B**), while maintaining train, validation, and test chromosomes, so that the test performance of these models could eventually be evaluated with new data (e.g., as we did with ATAC-seq and ChIP-seq from primary CD4$^+$ T cells, **Fig 6**). In addition to publishing the final maxATAC models, we took advantage of the good correlation between validation and test performance (**Figs 3G and S4H**) and mapped maxATAC scores to intuitive validation performance statistics precision, recall, $\log_2$(Precision / Precision$_{RANDOM}$), and F1-Score. Per TF model, the validation performance on Chr2 was averaged across each of the validation cell types. Given that precision is not necessarily a monotonic function of maxATAC score, we ensured one-to-one mapping in the following way: For a precision value mapping to multiple maxATAC scores, we selected the maxATAC score that maximized recall.

## maxATAC evaluation on scATAC-seq, primary cells and in discovery mode

**scATAC-seq.** We evaluated maxATAC on scATAC-seq during (i) method development (**S10D Fig**) and (ii) independent testing (**Fig 4**). For method development, we downloaded scATAC-seq fragment files for GM12878 (500, 1k, 5k, and 10k cell experiments) from the 10X Genomics website (https://www.10xgenomics.com/resources/datasets) [12]. For testing, we downloaded the high-loading, mixed-cell line scATAC-seq experiment SRX9633387 from GSE162690 [37], which we processed to cell-type specific pseudobulk fragment files.

Fragment files were filtered for reads aligning to the hg38 genome. For each pseudobulk, Tn5 cut sites were identified, signal tracks generated and minmax$_{99\%}$-normalized for maxATAC prediction, as described for bulk ATAC-seq.

To assess the impact of pseudobulk library size of TFBS prediction from maxATAC and TF motif scanning (**Figs 4E, 4F and S7**), we downsampled (*dplyr* (v1.0.8)) the GM12878 scATAC-seq library (10k cells, from the 10x Genomics website). Resulting down-sampled libraries were (1) processed for maxATAC prediction or (2) peak calling and motif scanning (see above).

**Primary cell types.** We downloaded ATAC-seq (GSE116696) and ChIP-seq data for 3 TFs (FOS:GSM3258569, MYC:GSM3258570, and JUNB:GSM3258571) in primary CD4$^+$ T cells stimulated with anti-CD3/anti-CD28 beads for 5 hours [46], processing ATAC-seq and ChIP-seq for maxATAC and precision-recall analysis as described above.

**Sequence-Specific TFBS prediction in CD4$^+$ T cells from atopic dermatitis patients.** We derived both DNA sequence and ATAC-seq signal inputs for maxATAC from a genetics atopic dermatitis (**AD**) study [54], combining whole genome sequencing (**WGS**) and OMNI-ATAC-seq (of peripheral blood CD4+ T-cells stimulated with anti-CD3 and anti-CD28 beads for 5 hours) for 6 AD patients and 6 age-matched controls. We used patient-specific genetic variant calls, focusing on the nine AD risk variants previously associated with allele-dependent chromatin accessibility [54]. We sought to identify pairs of AD patients and age-matched controls that were homozygous for the risk and non-risk alleles, respectively. Focusing on homozygous-risk and homozygous-nonrisk obviated the need for phasing (i.e., to resolve ATAC-seq signal into maternal and paternal DNA strands). We focused on two independent AD risk loci that met our criteria. For these loci, we applied maxATAC to patient- and control-specific DNA sequence and ATAC-seq signal to make TFBS predictions in the haplotype blocks containing the risk variants (linkage-disequilibrium $R^2 > .8$), for the patient pair, AD2 and CTL2. Haplotype blocks were defined by the homozygous risk variants identified above, and all SNPs in linkage disequilibrium with the risk variant ($R^2 \geq .8$) were used for patient-specific sequence prediction with maxATAC. The prediction window was defined by

the furthest genetic variant positions in LD with risk variant, extended by an additional + or -
512bp to contain the most distal variants in LD. We limited TFBS prediction to TFs with nom-
inal mRNA expression (DESeq2 VST counts $\geq$ 9) in activated CD4$^+$ T cells, measured in paral-
lel RNA-seq [54].

## Model interpretation

**Analysis of maxATAC model DNA sequence features.** We adapted model interpretation
tools DeepLIFT [43] and TF-MoDISco [44] to extract and cluster DNA sequence patterns rec-
ognized by the final GATA3 and CREM models. (Recall the "final" maxATAC models used all
available cell types for model training, but chromosomes 1 and 8 were excluded from model
building.) For each cell type independently, we analyzed inputs, centered on ChIP-seq peaks
from test chromosomes 1 and 8, and calculated importance scores keeping the ATAC-seq sig-
nal channel fixed. We examined the 15$^{th}$ bin out of the models' total 32-dimensional output
prediction array (corresponding to ChIP-seq peak centers).

## Supporting information

**S1 Table. Training data curated for 127 transcription-factor models from GEO and
ENCODE.** We tabulate the sample IDs, annotations and quality control metrics of the ChIP-
seq and ATAC-seq experiments, used to train the maxATAC models. In addition, we include
sample IDs excluded from the maxATAC training data, due to experimental perturbation.
(XLSX)

**S2 Table. Performance metrics for 74 transcription-factor models on bulk ATAC-seq.** Per-
formance metrics generated from maxATAC prediction for each cell-type and transcription-
factor pair. Results include test AUPR (chr1) for the following methods: maxATAC, TF motif
scanning, TOBIAS, Leopard, and average training ChIP-seq signal.
(XLSX)

**S3 Table. Test performance on scATAC-seq data for 193 cell type-transcription factor
combinations.** Results include test AUPR (chr1) for maxATAC models and TF motif scanning
on scATAC-seq and maxATAC models on bulk ATAC-seq, for 193 cell type-TF combina-
tions.
(XLSX)

**S4 Table. Atopic dermatitis risk loci with allele-dependent ATAC-seq signal.** Summary of
the variants associated with allele-dependent ATAC-seq signal and in linkage disequilibrium
with the (tagged) atopic dermatitis risk variant. Each genetic variant is annotated with the
patient genotype, Linkage Disequilibrium $R^2$ value, and gene expression information for genes
near these variants.
(XLSX)

**S5 Table. Extended maxATAC blacklist.** This BED file (hg38 coordinates) contains the black-
listed regions that were excluded from our analysis. These regions include the low mappability
arm of chr21, segmental duplications with high sequence similarity to chrM, telomeres, cen-
tromeres, and annotated gaps.
(XLSX)

**S1 Fig. maxATAC benchmark statistics. (A)** The number of unique TF models (y-axis) that
can be trained using different combinations of ENCODE, GEO, and in-house generated
OMNI ATAC-seq data (x-axis). TFs are broken into two categories: (1) those with only 2 cell
lines available (only cross-cell type training is feasible, red bar) and (2) models that have $\geq$ 3

cell lines (cross-cell type training *and* performance evaluation in a held-out test cell type are feasible, black bar). **(B)** The distribution of maxATAC TF models across TF families. The "unknown" category contains TFs that have not been associated with a TF family.
(TIF)

**S2 Fig. maxATAC "pan-cell-type, peak-centric" training approach. (A)** A small subset of our benchmark (11 TFs) was used to test peak-centric, pan-cell training (Methods) in a held-out cell type, GM12878. We varied the relative ratio of random and peak-centric regions ("Random Ratio") sampled during model training. Random ratio of 100% indicates that all training examples were randomly sampled from the genome, while 0% indicates that only peak-centric regions (centered on ChIP-seq and ATAC-seq peaks in the training cell type) were used. "Peak-centric" training enriches for TP examples and potentially challenging TN (high ATAC-seq signal but no TFBS). Furthermore,"pan-cell" training (red) additionally enriches for challenging TN examples, by pooling peak-centric regions across the training cell type. During pan-cell training, peak-centric regions are randomly sampled using signal from any of the training cell types (which may or may not have a TFBS in a given peak-centric region). This is in contrast to "cell type-specific" training (blue), in which peak-centric regions are specific to each of the training cell types and sampling of a peak-centric region is limited to signal from the cell types from which the peak-centric region was originally identified. **(B)** shows an example of 100% peak-centric, pan-cell training. (Note how peak-centric examples from HeLa come from both K562 and HeLa.) **(C)** The held-out test cell type, in addition to chr1 and chr8, are independent of model training and thereby enable performance evaluation of maxATAC model predictions in new cell types.
(TIF)

**S3 Fig. Schematic of precision-recall curve calculation for TFBS predictions with TF motifs.** This figure shows how a precision-recall curve is constructed from motif predictions and uses artificial data for a small genomic window. **(A)** For this example, the genome is divided into bins of length 200bp and sliding windows of length 200bp; windows overlapping blacklisted regions are removed. **(B)** ATAC-seq peaks are scanned with MOODS using the CisBP version 2 database of TF motifs. **(C)** TFBS predictions are ranked based on the number of motif occurrences within the bin. For TFs with >1 motif available, overlapping motif occurrences are counted as a single occurrence. True positive TFBS predictions are identified as those overlapping ChIP-seq peaks (yellow fill). **(D)** Bins are ranked by the number of TFBS prediction and compared to the gold standard. **(E)** Precision and recall are calculated at each threshold and summarized as a precision-recall curve.
(TIF)

**S4 Fig. Extended model performance statistics and comparisons. (A)** Comparison of median test precision at 5% recall between maxATAC TFBS predictions and TF motif scanning, where median is the median performance across each possible train-test cell type split. **(B-E)** Performance comparison of TOBIAS to maxATAC or simple TF motif-scanning TFBS prediction. Chromosome-wide test performance (chr1) using OMNI-ATAC of GM12878 (50k cells) [28] is reported for 60 TFs. TFs that do not have a known motif are highlighted in red. Test **(B)** AUPR or **(C)** precision at 5% recall of TOBIAS versus maxATAC. Test **(D)** AUPR or **(E)** precision at 5% recall of TOBIAS versus TF motif scanning in ATAC-seq peaks. **(F)** Comparison of median test precision at 5% recall between maxATAC TFBS predictions and TFBS prediction using averaged training ChIP-seq signal. **(G)** Test performance (precision at 5% recall) of maxATAC models compared to Leopard (DNase-seq-based) models using ATAC-seq input and maxATAC ChIP-seq gold standards for 8 cell lines and 7 TFs. maxATAC

outperforms Leopard for 17 out of 29 test performance comparisons. **(H)** Comparison of median test versus validation precision at 5% recall for maxATAC models constructed using all train-test cell type splits. Test corresponds to the test cell type and chromosome (chr1), while validation corresponds to training cell types and chromosome (chr2); $\rho = 0.91$ and $P < 10^{-15}$, n = 74 TFs. Validation **(I)** AUPR (median = 0.43) and **(J)** precision at 5% recall (median = 0.85) for the final 127 maxATAC TF models.
(TIF)

**S5 Fig. Factors contributing to maxATAC model performance. (A)** $AUPR_{MEDIAN}$ of maxATAC models as a function of number of cell types available for training. Validation performance (red) is estimated for the final maxATAC models (using all cell types available for model construction), n = 127 TFs, while test performance (blue) is available for 74 TF. **(B-C)** Factors contributing to relative test performance differences between maxATAC and average training ChIP-seq signal (74 TFs, analysis of median test AUPR). **(B)** $Log_2$-ratio of maxATAC AUPR relative to the AUPR of TFBS prediction using the averaged training ChIP-seq signal as a function of training cell types. **(C)** $Log_2$-fold-change, of maxATAC AUPR relative to the AUPR of TFBS prediction using training ChIP-seq signal, versus averaged Jaccard overlap between TFBS (from ChIP-seq) in pairs of training cell types.
(TIF)

**S6 Fig. Examples of cell-type-shared and -unique TF motifs learned by the maxATAC models.** For **(A)** GATA3 and **(B)** CREM maxATAC models, we modified TF-MoDISco to uncover importance-weighted DNA sequence patterns (summarized as CWM logos). TF-MoDISco was run independently on each of the training cell types, using positive TFBS examples (ChIP-centered inputs) on test chromosomes 1 and 8. ATAC signals for those regions were held constant and CWM were derived using the 15th bin of our 32-dimensional, 32bp-resolved TFBS prediction output, which corresponded to the center of the TF ChIP-seq peak. Percentages below each CWM indicate the fraction of positive examples containing the CWM in the 15th bin for each cell type.
(TIF)

**S7 Fig. The impact of scATAC-seq pseudobulk library size on TFBS prediction for maxATAC or TF motif scanning in accessible chromatin regions.** scATAC-seq libraries for GM12878 were down-sampled from 100M to 100k fragments, and test **(A)** AUPR and **(B)** precision at 5% recall performances evaluated. $Log_2(FC)$ indicates maxATAC performance relative to TF motif scanning for 60 TFs. Gray boxes indicate either (1) no known motif (e.g., GATAD2B, NFXL1, ZBTB40, ZNF207, ZNF592) or (2) no motif predictions for accessible chromatin regions detected at the given library size.
(TIF)

**S8 Fig. Experimental factors related to maxATAC performance on scATAC-seq and other ATAC-seq protocols. (A)** $Log_2(AUPR:AUPR_{mean\ per\ TF\ and\ TFBS\ method})$ normalized across ATAC-seq protocols per TF, for maxATAC and TF motif scanning separately. **(B)** The distribution of $log_2(AUPR:AUPR_{mean\ per\ TF\ and\ TFBS\ method})$. **(C)** Total reads across biological replicates (when available) for each experimental protocol. **(D)** Total reads that uniquely map across biological replicates. **(E)** Mean proportion of uniquely mapping reads across biological replicates per protocol.
(TIF)

**S9 Fig. An extended blacklist is needed for ATAC-seq. (A)** The maximum number of Tn5 cut site counts per autosomal chromosome for bulk OMNI-ATAC-seq (blue), 10x scATAC-

seq for 500 cells (yellow) and 5,000 cells (red) in GM12878. **(B)** IGV screenshot of a mitochondrial chromosome (chrM) segmentation duplication locus (green bar). While chromatin accessibility measurement by DNase-seq and scATAC-seq have no signal in this region, this is a region of extreme signal for OMNI-ATAC-seq, regardless of alignment methods. Pink tracks indicate OMNI-ATAC-seq mapped with multiple alignment methods (**Methods**). **(C)** IGV screenshot of the largely unmappable q-arm of chr21 and the relatively high signal caused by low mappability. Again, DNase-seq and scATAC-seq have no signal in this region, while OMNI-ATAC-seq has extreme signal, regardless of alignment strategy. These observations motivated our extended blacklist (**Methods**).
(TIF)

**S10 Fig. ATAC-seq-specific normalization is required to for robust prediction across ATAC-seq protocols. (A)** Max RP20M value per autosomal chromosome before and after filtering extended blacklist regions. **(B)** An example tandem repeat region with variable signal found on Chr17. **(C)** The *TRIM37* locus on Chr17 exhibits extreme, biologically relevant signal in the breast cancer cell line MCF-7 (red track). **(D)** Test AUPR in GM12878 for different alignment and normalization strategies (**Methods**) for bulk ATAC-seq or scATAC-seq (pseudobulk of 5k GM12878 cells). PCR duplicates and Tn5 cut sites that mapped to the extended blacklist (**Methods**) were removed prior to normalization. Test data is bulk ATAC-seq unless described as "scATAC". "95", "99", and "100" correspond to minmax normalization to the 95th, 99th or 100th-percentile highest ATAC-seq signal. "100" therefore corresponds to standard minmax normalization to the absolute max; this strategy was not robust to outlying ATAC-seq signal and therefore performs poorly when applied to different ATAC-seq alignment strategies or scATAC-seq. The far-right column represents performance on scATACseq data using (1) standard min-max normalization and (2) without applying the extended blacklist; this strategy has the worst performance generalizability to scATAC-seq.
(TIF)

**S11 Fig. Gene expression in AD patients and aged-matched controls. (A)** Median expression for each gene (DESeq2 VST-normalized counts) in the activated T cells RNA-seq dataset (6 AD patients and 6 age-matched controls). Blue dotted line indicates the nominal gene expression cutoff applied. Paired line plots showing the difference in gene expression between AD patients and their age-matched controls for **(B)** *TNFRSF6B*, **(C)** *RTEL1-TNFRSF6B*, **(D)** *RTEL1*, **(E)** *LIME1*, **(F)** *SLC2A4RG*, and **(G)** *ZGPAT*. Each point is colored according to whether the donor was homozygous risk (red), heterozygous risk (yellow), and homozygous non-risk (green). Each line is colored according to the donor pair.
(TIF)

**S12 Fig. maxATAC predictions for 105 TFs in AD2 and CTL2.** Heatmaps show TFBS predictions (32bp width) for **(A)** CTL2 and **(B)** AD2, using a score cutoff that maximizes the average F1-score across validation cell types. **(C)** Differential TFBS between AD2 and CTL2, where red indicates AD2-specific prediction, blue indicates CTL2-specific prediction and grey denotes no difference.
(TIF)

## Acknowledgments

We thank several sources of computational support for this work: the Biomedical Informatics Research IT Group at Cincinnati Children's Hospital Medical Center (P. Velayutham), the Ohio Supercomputer Center and the Advanced Research Computing Center at the University

of Cincinnati Research. We acknowledge Christopher Benner, Shaun Mahony, Ivan Marazzi, Brad Rosenberg, Krishna Roskin and Andrea Toth for manuscript feedback and Wen Niu for help cross-referencing ChIP-seq experiments across databases. Some of our images were created with BioRender.com (Figs 1, 6 and 7) and Lucidchart (www.lucidchart.com) (Figs 2 and S3).

## Author Contributions

**Conceptualization:** Emily R. Miraldi.

**Data curation:** Tareian A. Cazares, Faiz W. Rizvi, Michael Kotliar, Sreeja Parameswaran, Leah C. Kottyan, Matthew T. Weirauch, Emily R. Miraldi.

**Funding acquisition:** Leah C. Kottyan, Artem Barski, Matthew T. Weirauch, V. B. Surya Prasath, Emily R. Miraldi.

**Investigation:** Tareian A. Cazares, Faiz W. Rizvi, Balaji Iyer, Xiaoting Chen, Anthony T. Bejjani, Joseph A. Wayman, Omer Donmez, Benjamin Wronowski.

**Methodology:** Tareian A. Cazares, Faiz W. Rizvi, Balaji Iyer, Xiaoting Chen, Joseph A. Wayman, V. B. Surya Prasath, Emily R. Miraldi.

**Project administration:** Tareian A. Cazares, Emily R. Miraldi.

**Software:** Tareian A. Cazares, Faiz W. Rizvi, Balaji Iyer, Xiaoting Chen, Michael Kotliar, Emily R. Miraldi.

**Supervision:** Leah C. Kottyan, Artem Barski, Matthew T. Weirauch, V. B. Surya Prasath, Emily R. Miraldi.

**Validation:** Tareian A. Cazares, Faiz W. Rizvi, Balaji Iyer, Xiaoting Chen, Anthony T. Bejjani.

**Visualization:** Tareian A. Cazares, Faiz W. Rizvi, Xiaoting Chen, Emily R. Miraldi.

**Writing – original draft:** Tareian A. Cazares, Faiz W. Rizvi, Emily R. Miraldi.

**Writing – review & editing:** Tareian A. Cazares, Faiz W. Rizvi, Xiaoting Chen, Joseph A. Wayman, Leah C. Kottyan, Artem Barski, Matthew T. Weirauch, V. B. Surya Prasath, Emily R. Miraldi.

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
