## [Decision Letter · Decision Letter 0]

12 Sep 2022

Dear Miraldi,

Thank you very much for submitting your manuscript "maxATAC: genome-scale transcription-factor binding prediction from ATAC-seq with deep neural networks" for consideration at PLOS Computational Biology.

As with all papers reviewed by the journal, your manuscript was reviewed by members of the editorial board and by several independent reviewers. In light of the reviews (below this email), we would like to invite the resubmission of a significantly-revised version that takes into account the reviewers' comments.

We cannot make any decision about publication until we have seen the revised manuscript and your response to the reviewers' comments. Your revised manuscript is also likely to be sent to reviewers for further evaluation.

Sincerely,

Teresa M. Przytycka

Academic Editor

PLOS Computational Biology

William Noble

Section Editor

PLOS Computational Biology

Reviewer's Responses to Questions

**Comments to the Authors:**

Reviewer #1: The authors have presented a computational framework (maxATAC) based on deep convolutional neural networks for predicting transcription factor (TF) binding from ATAC-seq profiles. Authors demonstrated TF binding side prediction from ATAC-seq in new cell types. For benchmarking, authors curated an extensive dataset of existing cell-type specific ChIP-seq and ATAC-seq datasets. Data were manually verified using annotations of each experiment. Their models performed well on both ATAC-seq and scATAC-seq. Quality-controlled, processed datasets are also available to download. Overall, it is a well-written manuscript with an apt description of the method.

Minor comment

The deep learning approach used in this work is not considered as novel (e.g. Tianqi Yang et al, Bioinformatics 2022, Laiyi Fu et al, Science Advances 2020). While this work is unique on capability of trans-cell TFBS predictions and providing community access for quality-controlled, processed, ready-to-use curated large dataset to advance gene regulatory network research on ATAC-seq data

Reviewer #2: This manuscript curated a good quality dataset for ATAC-seq and TF ChIP-seq., and present a method named maxATAC for prediction of TF binding sites. TF binding site prediction is an important problem in gene regulatory analysis, and has broad application in many fields. The curated dataset will be useful for future method developments. The manuscript writing is clear and easy to follow. However, I have one concern about the validation of the method.

1. There are several methods available for TF binding site prediction from chromatin accessibility data, for example methods presented in the ENCODE-DREAM challenge. Using different data as input and comparing AUPR is not fair. The reason authors did not compare maxATAC with those method is the difference of input data. I agree that there are some differences between ATAC and DNase. However, for most case, these two data are highly correlated. Authors can train those methods on the curated data they collected and perform direct comparison with those methods.

2. To evaluate the motif scanning, how authors calculate the AUPR is not clear. I assuming they slide the motif matching score to calculate the AUPR. Another way to perform comparison is sliding sum of (or product of) motif matching score and ATAC-seq signal (RPKM, or openness) to calculate AUPR.

Reviewer #3: The Cazares et al. manuscript presents a well-curated benchmark dataset that pairs ATAC-seq and published transcription factor (TF) ChIP-seq in 20 cell types for an unprecedented number of TFs (127 TFs with ChIP-seq data in at least 2 cell types; 74 TFs of them with data in at least 3 cell types). The authors generated their own OMNI-ATAC-seq for some cell lines to assemble the resource. In addition to the dataset, the paper proposes a deep learning model that predicts trans-cell-type TF occupancy based on a dilated CNN model. The maxATAC model was carefully evaluated using both bulk and single-cell ATAC-seq data on various cell types and TFs in a held-out chromosome and held-out cell type manner and was shown to reach comparable (though perhaps slightly weaker) performance to several current state-of-the-art methods for cross-cell-type TF occupancy prediction developed for DNase-seq. The maxATAC approach also had superior performance compared to traditional motif scanning or ChIP-seq signal averaging across training cell types for most TFs.

Overall, this manuscript is well-written and results are nicely presented, with the limitations of the study helpfully described in the Discussion. While there is limited technical or conceptual novelty relative to previous neural network models that tackle the same or related problems, the benchmark dataset establishes a useful resource, and the maxATAC performance results lay down a useful baseline for future deep learning methods. More exploration of the central question of cross-cell-type generalizability of TF occupancy prediction, wider method comparison, and better interpretation of the trained models would strengthen the paper.

Major points:

1. The key question for the paper is the extent of generalizability of TF binding models to new cell types, where potentially new co-factors or members of the TF complex may be expressed and alter the sequence recognition code. The authors present various analyses to explain the variability in maxATAC’s performance over TFs, but the issue of the cell-type-specificity of the underlying sequence signal is not fully developed. For example, in Figure S5, the authors show the maxATAC auPR relative to training ChIP-seq signal auPR (as a log odds ratio) vs Jaccard distance over the training ChIP-seq samples. This tells us that when a factor binds nearly the same sites in all cell types (e.g. CTCF), it is hard to outperform training ChIP-seq signal, whereas the model can outperform this baseline when there are cell-type-specific sites. However, it absolute terms, maxATAC has the highest auPR on CTCF test data out of all factors, because it also has a highly conserved binding motif across cell types. So it would be helpful to address how the model performs on unseen cell types for TF with cell-type-specific binding motifs as compared to highly conserved binding motifs.

2. As a related issue, a model interpretation analysis, e.g. using feature attribution to identify cell-type-specific vs. conserved motifs, might also help investigating the issue of generalizability. In particular, is the model actually learning co-factor binding signals in addition to the TF motif? Can the authors show that the model is learning multiple modes of binding for TFs with cell-type-specific binding motifs/patterns?

3. The major method comparison was made between maxATAC and two baseline methods (motif-scanning and ChIP-seq average signal), but since there are also some published ATAC-seq based TF “footprinting” approaches such as HINT-ATAC and TOBIAS, i.e. methods that try to model the ATAC-seq signal to better local the potential TF binding site, it would be interesting to see a comparison with them. It would also be interesting to understand whether the maxATAC model is learning a “footprinting” method (relatively protected region within a peak) or simply finding peak summits.

4. It would also be interesting to compare the deep learning method to “shallow learning”, e.g. a kernel method like gkm-SVM combined with a simple kernel on ATAC-seq signal.

5. There are also some improvements that could be made to make the presentation of this paper clearer. Various model and training details (explicit explanation of the output of the model, ground truth used for training, details of the encoding and 32bp resolution) are not clear until the methods, and some details are obscure even there? It would be helpful to include a figure and overview of the model in the main text. The training set-up might be described in the ENCODE-DREAM challenge, but a self-contained presentation would be helpful here. Also, the wording “a suite of models” and “a collections of models” is a bit misleading – it is not true that there is collection of models with different structures, as is finally clarified in the model description section, but rather there are models for different TFs (all with the same architecture).

Minor points:

- Fig S1A: should have a consistent use of cell-line-specific and cell-type-specific in the figure caption

- Line 198: should be sequence-based

**Have the authors made all data and (if applicable) computational code underlying the findings in their manuscript fully available?**

Reviewer #1: Yes

Reviewer #2: Yes

Reviewer #3: Yes

PLOS authors have the option to publish the peer review history of their article (what does this mean?). If published, this will include your full peer review and any attached files.

Reviewer #1: No

Reviewer #2: No

Reviewer #3: No
---

## [Decision Letter · Decision Letter 1]

10 Jan 2023

Dear Miraldi,

We are pleased to inform you that your manuscript 'maxATAC: genome-scale transcription-factor binding prediction from ATAC-seq with deep neural networks' has been provisionally accepted for publication in PLOS Computational Biology.

Best regards,

Teresa M. Przytycka

Academic Editor

PLOS Computational Biology

William Noble

Section Editor

PLOS Computational Biology

Reviewer's Responses to Questions

**Comments to the Authors:**

Reviewer #1: Authors addressed my comments

Reviewer #2: I do not have further comments.

Reviewer #3: The revised paper has sufficiently addressed most of the issues described in the major and minor comments of the previous review.

For comments 1 and 2, the authors have added model interpretation results using TFMoDISco, which provides an interesting investigation of some limitations in cross-cell-type generalization due to cell-type-specific binding modes. Note that it might be helpful to highlight GATA3 and CREM in Fig. 5C for better visibility. Comment 3 has to some extent been addressed with the comparison to TOBIAS and the neural network activation maximization plot, and the authors suggest that the model is indeed learning a “footprint” region (although hard to disambiguate from simply a peak summit region). The authors mention two papers (DeepSEA and a 2016 paper from the Gifford lab) to justify not performing the comparison with gkm-SVM in comment 4. This is fine, though please note that there are issues with some of the early CNN literature in genomics; for example, the original DeepSEA results may largely reflect GC content bias, and later papers regress out GC content prior to training models. For comment 5, the presentation of the maxATAC methodology is now much clearer with the additional information provided in Figure 2.

**Have the authors made all data and (if applicable) computational code underlying the findings in their manuscript fully available?**

Reviewer #1: Yes

Reviewer #2: Yes

Reviewer #3: Yes

PLOS authors have the option to publish the peer review history of their article (what does this mean?). If published, this will include your full peer review and any attached files.

Reviewer #1: **Yes: **Hatice Ulku Osmanbeyoglu

Reviewer #2: No

Reviewer #3: No

---

## [Editor Report · Acceptance letter]

26 Jan 2023

PCOMPBIOL-D-22-01037R1 

maxATAC: genome-scale transcription-factor binding prediction from ATAC-seq with deep neural networks

Dear Dr Miraldi,

I am pleased to inform you that your manuscript has been formally accepted for publication in PLOS Computational Biology. Your manuscript is now with our production department and you will be notified of the publication date in due course.

With kind regards,

Zsofi Zombor
